# AudioX: A Unified Framework for Anything-to-Audio Generation

**Zeyue Tian[1], Zhaoyang Liu[1], Yizhu Jin[1], Ruibin Yuan[1], Liumeng Xue[1],**
**Xu Tan[2], Qifeng Chen[1], Wei Xue[†1], Yike Guo[†1]**

[1]Hong Kong University of Science and Technology
[2]Independent Researcher

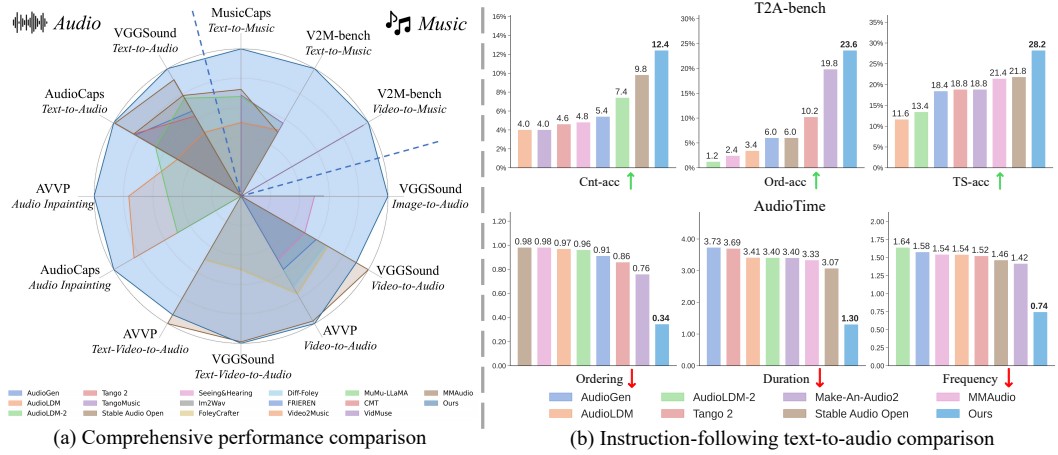

(a) Comprehensive performance comparison  (b) Instruction-following text-to-audio comparison

Figure 1: **Performance comparison of AudioX against baselines.** (a) Comprehensive comparison across multiple benchmarks via Inception Score. (b) Results on instruction-following benchmarks.

## Abstract

Audio and music generation based on flexible multimodal control signals is a widely applicable topic, with the following key challenges: 1) a unified multimodal modeling framework, and 2) large-scale, high-quality training data. As such, we propose AudioX, a unified framework for anything-to-audio generation that integrates varied multimodal conditions (*i.e.*, text, video, and audio signals) in this work. The core design in this framework is a Multimodal Adaptive Fusion module, which enables the effective fusion of diverse multimodal inputs, enhancing cross-modal alignment and improving overall generation quality. To train this unified model, we construct a large-scale, high-quality dataset, IF-caps, comprising over 7 million samples curated through a structured data annotation pipeline. This dataset provides comprehensive supervision for multimodal-conditioned audio generation. We benchmark AudioX against state-of-the-art methods across a wide range of tasks, finding that our model achieves superior performance, especially in text-to-audio and text-to-music generation. These results demonstrate our method is capable of audio generation under multimodal control signals, showing powerful instruction-following potential. The code and datasets will be available at `https://zeyuet.github.io/AudioX/`.

---

[†]Corresponding Authors

# 1 INTRODUCTION

In recent years, audio generation, especially for sound effects and music, has emerged as a crucial element in multimedia creation, showing practical values in enhancing user experiences across a wide range of applications. For example, in social media, film production, and video games, sound effects and music significantly intensify emotional resonance and engagement with the audience. The ability to create high-quality audio not only enriches multimedia content but also opens up new avenues for creative expression.

However, the manual production of audio is time-consuming and requires specialized skills, presenting a compelling research opportunity to automate audio generation. Despite notable advancements (Liu et al., 2023; Copet et al., 2024; Wang et al., 2024), the field has predominantly focused on specialized models with constrained inputs and outputs. These models often operate with a single conditioning modality, such as text-to-audio or video-to-audio, and are typically restricted to a single output domain, like generating either sound effects (Cheng et al., 2025) or music (Tian et al., 2025) exclusively. While a recent trend towards unification is emerging, with some pioneering works accommodating multiple inputs (Polyak et al., 2024; Zhang et al., 2024), they often lack the flexibility to support diverse modal combinations and exhibit weak instruction-following abilities. As a result, the potential of unified models still remains underexplored. We find that a major factor behind these limitations is the scarcity of high-quality, multimodal data suitable for training unified systems. Existing datasets are often task-specific, typically providing supervision for only one conditioning modality, such as text-to-audio (Kim et al., 2019), video-to-audio (Chen et al., 2020), or video-to-music (Tian et al., 2025). This lack of datasets with diverse and combinable control signals has significantly hindered the development and training of unified models.

To this end, we propose a unified framework termed AudioX for anything-to-audio generation. We observe that Transformer-based works (Wu et al., 2023a; Liu et al., 2024b; Lin et al., 2023) have effectively tackled multi-modal alignment, and we build on this success by incorporating Transformer-based methods into our framework for multi-modal condition handling. Furthermore, diffusion models have increasingly become leading-edge techniques in the field of high-quality audio and music generation (Evans et al., 2024a;b), outperforming next-token prediction in terms of audio fidelity (Evans et al., 2024a; Majumder et al., 2024). Therefore, we mainly build on Diffusion Transformer (DiT) to unify multimodal conditions and generate high-fidelity audio. To further enhance multimodal representation learning and alignment, we introduce a lightweight Multimodal Adaptive Fusion module that adaptively weights and aligns conditioning modalities before fusion, enabling stronger cross-modal control and yielding significant improvements in generation quality.

To support the training of a unified model, we designed a pipeline using structured annotation and data augmentation to build IF-caps (**I**nstruction-**F**ollowing), a large-scale, high-quality multimodal dataset. The dataset serves as a robust foundation for our approach, containing over 1.3 million general audio samples and 5.7 million music samples. Training on this large-scale, fine-grained dataset allows our model to handle flexible multimodal conditions and generate diverse audio genres, including music and sound effects. Consequently, AudioX enables a range of tasks, including text-to-audio generation, video-to-audio generation, audio inpainting, and text-guided music completion.

With this unified design and trained on our large-scale dataset, our model demonstrates exceptional performance and strong instruction-following capabilities. To validate our model's capabilities, we benchmark it against state-of-the-art methods across a comprehensive suite of tasks and established benchmarks. In addition, to rigorously evaluate its instruction-following ability on T2A tasks, we construct a new benchmark, T2A-bench. As demonstrated in Sec. 5.3, AudioX achieves state-of-the-art or comparable results across multiple benchmarks and various tasks, while substantially outperforming prior methods in instruction-following capabilities. A notable finding from our unified training approach is that we observe a ***cross-modal regularization effect*** under unified training: improving the quality and granularity of textual supervision reduces alignment noise and leads to better modality alignment, which jointly improves performance across conditioning modalities (see Sec. 5.4). This observation provides empirical insight for future multimodal audio generation.

In summary, the main contributions of this work are as follows: 1) We propose AudioX, a unified framework for anything-to-audio generation that overcomes the limitations of constrained inputs and outputs. The proposed framework supports audio and music generation from varied multi-modal conditions, contributing to a new insight into studying generalist models for audio generation.

2) To overcome data scarcity for unified training, we design a data curation pipeline and construct a large-scale, high-quality dataset, IF-caps, containing over 7 million samples with fine-grained annotations.

3) We conduct comprehensive experiments on a wide array of tasks, systematically benchmarking state-of-the-art methods categorized by their input modalities and output domains. Our extensive experiments demonstrate our model's strong multi-task capabilities and superior instruction-following ability.

## 2 RELATED WORK

**Audio and music generation.** Recent advances in deep generative models have greatly broadened the scope of audio and music synthesis. However, most existing methods remain confined to a single modality or support only limited types of conditioning. For instance, *text-to-audio* approaches (Liu et al., 2023; Majumder et al., 2024; Evans et al., 2024a;b; Jiang et al., 2025; Huang et al., 2023; He et al., 2024) focus on generating diverse soundscapes from textual prompts, while *text-to-music* systems (Copet et al., 2024; Ghosal et al., 2023; Deng et al., 2024; Yuan et al., 2024; 2025; Ma et al., 2024) specialize in composing coherent musical pieces. Separate lines of work tackle tasks like *audio inpainting* (Liu et al., 2023; 2024a), primarily with text conditioning. Meanwhile, *video-to-audio* methods (Zhang et al., 2024; Luo et al., 2024; Wang et al., 2024; Polyak et al., 2024; Chen et al., 2024) typically generate foley or environmental sounds synchronized to visual cues. Some of these also incorporate text for additional context, thereby bridging visual and textual modalities. Beyond sound effects, *video-to-music* approaches (Kang et al., 2024; Liu et al., 2024c; Di et al., 2021; Tian et al., 2025; Li et al., 2024b; Lin et al., 2024; Li et al., 2024a) align musical compositions with the visual content to enhance narrative depth in multimedia applications. Despite these advances, current frameworks often specialize in only one modality or rely on a limited set of input conditions, hindering multi-task adaptation and restricting their ability to scale or transfer knowledge across related tasks. In contrast, our *unified* approach supports both audio and music generation for a broad range of input conditions—including text, video, and audio—all within a single framework.

**Audio Datasets.** While substantial research efforts have led to the creation of valuable datasets for specific tasks like text-to-audio (Kim et al., 2019; Xue et al., 2025; Drossos et al., 2020; Wu et al., 2023b), text-to-music (Copet et al., 2024; Liu et al., 2024c; Ramires et al., 2020), video-to-audio (Chen et al., 2020; Hershey et al., 2021; Tian et al., 2020), and video-to-music (Tian et al., 2025; Zhou et al., 2025; Chi et al., 2024), their utility for training a generalist unified model remains limited. These resources are typically constrained to a single conditioning modality and a narrow output domain (e.g., only sound effects or only music). This fragmentation of data has significantly hindered progress towards developing more versatile and robust systems. To overcome this critical data scarcity, we introduce a large-scale, multimodal dataset constructed via a novel annotation and augmentation pipeline, specifically designed to provide the comprehensive supervision required for unified audio and music generation.

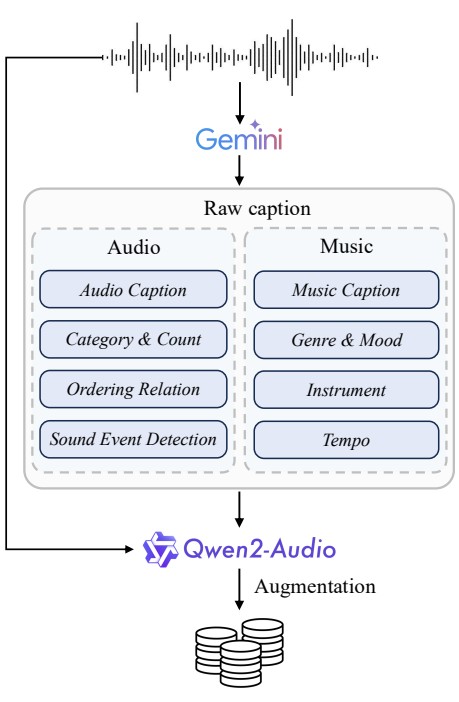

Figure 2: Dataset process pipeline.

**Diffusion models.** Denoising diffusion models (Ho et al., 2020; Song et al., 2020) have become a cornerstone of modern generative modeling, achieving state-of-the-art results in image (Rombach et al., 2022; Ramesh et al., 2022; Brooks et al., 2023), video (Chen et al., 2023; Ho et al., 2022; Guo et al., 2023; He et al., 2024), and audio synthesis (Popov et al., 2021; Jeong et al., 2021; Liu et al., 2024a; 2023; 2022; Evans et al., 2024a). However, their

application in the audio domain has predominantly been limited to single-condition tasks (e.g., text-to-audio), falling short of the more generalized "anything-to-audio" scenarios where inputs can be multimodal. To bridge this gap, our unified framework leverages the power of diffusion models for multi-condition generation, offering a more flexible and universal solution.

## 3   DATASET PROCESS

Existing audio datasets often lack the high-quality, multimodal conditioning signals necessary to train versatile, unified models. To address this gap, we designed an effective annotation pipeline that processes existing video datasets (Chen et al., 2020; Hershey et al., 2021; Tian et al., 2025), allowing us to construct IF-caps, a large-scale dataset with diverse, multi-modal conditions. Our pipeline, as shown in Fig. 2, operates as follows: ***First***, we employ a powerful multimodal LLM (Gemini 2.5 Pro) to generate a comprehensive set of initial annotations by processing the audio track of each 10-second video-audio clip. These annotations consist of a holistic global caption and a set of structured fields. For general audio, these fields include sound event classification and count; for music, they specify attributes like genre and instrumentation. ***Then***, since using the resource-intensive Gemini model for the entire dataset is costly, we leverage the open-source Qwen2-Audio (Chu et al., 2024) model to augment these structured fields at a large scale. Conditioned on both the initial annotations and the raw audio, the model generates varied captions, enhancing data diversity while managing costs. ***Finally***, this process yields comprehensive, fine-grained captions for approximately 1.3 million video-audio clips and 5.7 million video-music clips. The diversity of our curated dataset is highlighted by the word clouds in Fig. 3. More details and samples of our annotated data are provided in the Appendix A.1.2.

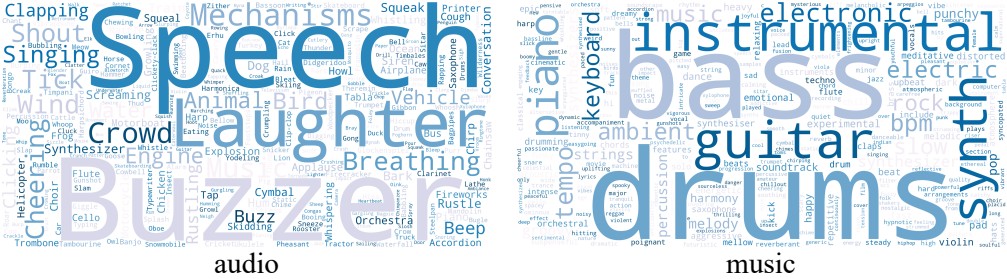

Figure 3: Word clouds for our curated dataset, illustrating the diversity of terms for the general audio (left) and music (right) domains.

## 4   METHOD

### 4.1   MODEL DESIGN

Our framework, AudioX, as shown in Fig. 4, is built upon a DiT backbone designed for high-fidelity audio synthesis. Given video $\mathbf{X}_v$, text $\mathbf{X}_t$, and audio $\mathbf{X}_a$, each modality is passed through corresponding specialized encoders. To capture the temporal dynamics, the resulting video and audio features are then processed by a temporal transformer. Finally, the features from all three modalities are mapped through a projection head to produce the domain-specific embeddings ($\mathbf{H}_v$, $\mathbf{H}_t$, $\mathbf{H}_a$). These embeddings are then fused into a unified condition embedding, which is ultimately passed to the Diffusion Transformer to guide the generation process.

A key challenge in training a unified model is that signals from different modalities can interfere with each other, making effective fusion and well-aligned conditioning critical. To address this, we introduce the lightweight **Multimodal Adaptive Fusion (MAF)** module. As shown in Fig. 4 (right), the MAF module operates as follows: *First*, the initial feature embeddings from each modality are fed into *gates*, which filter and reweight them to suppress noise and retain the most informative cues. *Next*, the gated embeddings are concatenated and attended by *learnable queries* via cross-attention. These queries are organized into three modality-specific sets, acting as experts to assess and aggregate evidence across the different data streams. *Finally*, a *self-attention* layer consolidates this

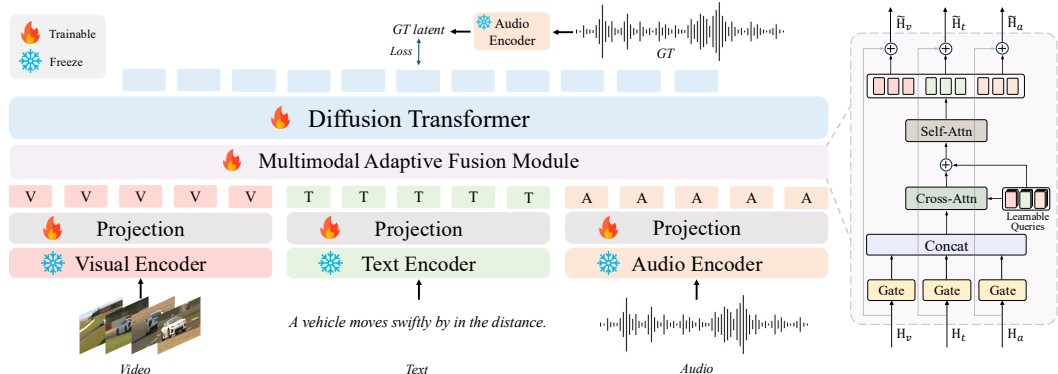

Figure 4: **The AudioX Framework.** Specialized encoders process diverse modalities, and a MAF module unifies these signals into a conditioning embedding $H_c$. The DiT backbone processes the noisy latent input $z_t$, conditioning on $H_c$ via cross-attention to generate high-quality audio and music. ($z_t$ and $H_c$ notations are omitted for visual clarity).

aggregated context, and the refined information is dispatched back to the modality paths via residual updates. This process yields calibrated, modality-specific outputs which are then concatenated to form the final multimodal condition embedding, $\mathbf{H}_c$:

$$\tilde{\mathbf{H}}_v,\ \tilde{\mathbf{H}}_t,\ \tilde{\mathbf{H}}_a = \mathrm{MAF}(\mathbf{H}_v,\ \mathbf{H}_t,\ \mathbf{H}_a),\quad \mathbf{H}_c = \mathrm{Concat}\!\left(\tilde{\mathbf{H}}_v,\ \tilde{\mathbf{H}}_t,\ \tilde{\mathbf{H}}_a\right). \tag{1}$$

This final embedding, along with a diffusion timestep $t$, is what conditions the DiT backbone for the final audio synthesis. As we demonstrate in our ablation studies (Sec. 5.4), the MAF module is essential for reducing cross-modal interference while improving both the overall generation quality on multimodal tasks and the model's instruction-following capabilities.

## 4.2 TRAINING

The objective of the training process is to effectively integrate multimodal inputs and optimize the DiT model for generating high-quality audio or music through a robust diffusion and denoising framework. The details of the training data are provided in Table A.1 in the Appendix. During training, for each pair $(\mathbf{X}_v, \mathbf{X}_t, \mathbf{X}_a \mid \mathbf{A})$, where $\mathbf{A}$ is the ground truth we aim to generate, if the pair lacks video or audio modality input, we use zero-padding to fill the missing modality. If it lacks text modality input, we substitute with natural language descriptions, such as "Generate music for the video." for the video-to-music generation task. For the tasks of audio inpainting and music completion, the audio modality input is required. In audio inpainting, $\mathbf{X}_a$ is a masked version of the ground truth audio $\mathbf{A}$, and the model's objective is to fill in the masked sections. For music completion, $\mathbf{X}_a$ is the preceding music segment of $\mathbf{A}$, and the model aims to generate the subsequent music segment of $\mathbf{X}_a$.

**Diffusion process.** The DiT model processes the multimodal embedding $\mathbf{H}_c$ in the latent space through a denoising diffusion process. Initially, the ground truth $\mathbf{A}$ is encoded using an encoder $\mathcal{E}$, which projects $\mathbf{A}$ into the latent space, yielding the latent representation $\mathbf{z} = \mathcal{E}(\mathbf{A})$. The data then undergoes a forward diffusion process, producing noisy latent states at each timestep $t$.

The forward diffusion is defined as a Markov process over $T$ timesteps, where the latent state at timestep $t$ is produced based on the latent state at $t-1$:

$$q(\mathbf{z}_t|\mathbf{z}_{t-1}) = \mathcal{N}(\mathbf{z}_t; \sqrt{1-\beta_t}\mathbf{z}_{t-1}, \beta_t\mathbf{I}), \tag{2}$$

where $\beta_t$ represents the predefined variance at timestep $t$, and $\mathcal{N}$ denotes a Gaussian distribution. The forward diffusion process gradually adds noise to the latent state.

The reverse denoising process involves training a transformer network $\epsilon_\theta$ to gradually remove noise at each timestep and reconstruct the clean data. The reverse process is modeled as follows:

$$p_\theta\left(\mathbf{z}_{t-1}|\mathbf{z}_t\right) = \mathcal{N}\left(\mathbf{z}_{t-1}; \mu_\theta\left(\mathbf{z}_t, t, \mathbf{H}_c\right), \mathbf{\Sigma}_\theta\left(\mathbf{z}_t, t, \mathbf{H}_c\right)\right), \qquad (3)$$

where $\mu_\theta$ and $\mathbf{\Sigma}_\theta$ are the predicted mean and covariance of the reverse diffusion, conditioned on $\mathbf{z}_t$, $t$, and $\mathbf{H}_c$. These parameters define the Gaussian distribution from which $\mathbf{z}_{t-1}$ is sampled.

The denoiser network $\epsilon_\theta$ takes as input the noisy latent state $\mathbf{z}_t$, timestep $t$, and the multimodal condition embedding $\mathbf{H}_c$. The goal is to minimize the noise estimation error at each timestep, which is formulated as:

$$\min_\theta \mathbb{E}_{t,\mathbf{z}_t,\epsilon} \left\| \epsilon - \epsilon_\theta\left(\mathbf{z}_t, t, \mathbf{H}_c\right) \right\|_2^2, \qquad (4)$$

where $\epsilon$ is the simulated noise at timestep $t$, and $\epsilon_\theta(\mathbf{z}_t, t, \mathbf{H}_c)$ is the predicted noise from the model. The training objective is to minimize the mean squared error between the simulated and predicted noise across all timesteps.

By training the DiT model in this manner, we effectively unify multimodal inputs into a latent space, enabling the generation of high-quality audio or music that is coherent and aligned with the input conditions.

## 5 EXPERIMENTS

In this section, we provide the implementation details of our experiments and conduct extensive evaluations. These assessments comprehensively measure the effectiveness of our proposed method from both subjective and objective viewpoints. The evaluations aim to offer valuable insights into the generation of audio and music from various inputs.

### 5.1 IMPLEMENTATION DETAILS

For encoding the visual features, we use CLIP-ViT-B/32 (Radford et al., 2021) to extract video frame features at a rate of 5 fps, and Synchformer (Iashin et al., 2024) to extract synchronization features at 25 fps. The CLIP and Synchformer features are fused via addition. The text inputs are encoded using T5-base (Raffel et al., 2020), while the audio is encoded and decoded using an audio Autoencoder (Evans et al., 2024b). The model has a total of 2.4B parameters (1.1B trainable). Our proposed MAF module constitutes only 60M of these parameters, highlighting its lightweight nature. The DiT model, consisting of 24 layers, uses a pretrained model from (Evans et al., 2024b).

The training process uses the AdamW optimizer with a base learning rate of 1e-5, weight decay of 0.001, and a learning rate scheduler incorporating exponential ramp-up and decay phases. To improve inference stability, we maintain an exponential moving average of the model weights. Training is conducted on three clusters of NVIDIA H800 GPUs, each with 80GB of memory, requiring approximately 4k GPU hours in total. The batch size is set to 48. During inference, we perform 250 steps using classifier-free guidance with a scale of 7.0. Please refer to Appendix A.1.1 for further details on our training and evaluation datasets.

### 5.2 EVALUATION METRICS

To provide a comprehensive assessment of our model, we employ a suite of objective and subjective metrics. Further details for each metric are provided in the Appendix A.2.

**Objective Evaluation.** For overall audio quality and semantic alignment, we use several established metrics. These include: Kullback-Leibler Divergence (KL); Inception Score (IS); Fréchet Distance (FD) with PANNs embeddings (Kong et al., 2020); Fréchet Audio Distance (FAD) with VGGish embeddings (Hershey et al., 2017); Production Complexity (PC) and Production Quality (PQ) (Tjandra et al., 2025). As a prompt-free metric for both quality and diversity, we chose IS for the unified comparison in Fig. 1. For alignment, we use the CLAP score (Wu et al., 2023b) for text inputs and the Imagebind AV score (Girdhar et al., 2023) for video inputs. To assess the model's instruction-following capabilities in T2A, we report metrics on two benchmarks. On our proposed

Table 1: **Performance evaluation across various tasks and datasets.** Task abbreviations are: T2A (Text-to-Audio), V2A (Video-to-Audio), TV2A (Text-and-Video-to-Audio), T2M (Text-to-Music), V2M (Video-to-Music), and TV2M (Text-and-Video-to-Music). For alignment (Align.), we use the CLAP score for text and the Imagebind AV score for video inputs.

| Dataset | Method | Task | KL ↓ | IS ↑ | FD ↓ | FAD ↓ | PC ↑ | PQ ↑ | Align. ↑ |
|---|---|---|---|---|---|---|---|---|---|
| AudioCaps | AudioGen(Kreuk et al., 2022) | T2A | 1.39 | 10.22 | 13.29 | 1.72 | 3.26 | 5.25 | 0.27 |
| | AudioLDM-L-Full(Liu et al., 2023) | T2A | 2.00 | 6.51 | 37.27 | 8.37 | 2.82 | 5.67 | 0.20 |
| | AudioLDM-2-Large(Liu et al., 2024a) | T2A | 1.49 | 8.46 | 26.34 | 1.97 | 2.86 | 5.77 | 0.22 |
| | Tango 2(Majumder et al., 2024) | T2A | **1.11** | 10.37 | 12.22 | 3.20 | **3.63** | 5.82 | **0.36** |
| | Stable Audio Open(Evans et al., 2024b) | T2A | 2.01 | 10.37 | 29.01 | 3.15 | 2.77 | **6.16** | 0.21 |
| | MAGNET-large(Ziv et al., 2024) | T2A | 1.62 | 7.46 | 24.88 | 2.99 | 3.25 | 5.15 | 0.15 |
| | MMAudio(Cheng et al., 2025) | T2A | 1.35 | 12.03 | 12.63 | 4.71 | 3.06 | 5.64 | 0.30 |
| | AudioX | T2A | 1.27 | **12.48** | 11.51 | 1.59 | 3.32 | 5.80 | 0.30 |
| VGGSound | AudioGen(Kreuk et al., 2022) | T2A | 2.16 | 11.09 | 15.94 | 2.48 | 3.30 | 5.45 | 0.29 |
| | AudioLDM-L-Full(Liu et al., 2023) | T2A | 2.41 | 6.52 | 31.15 | 7.05 | 2.93 | 5.99 | 0.27 |
| | AudioLDM-2-Large(Liu et al., 2024a) | T2A | 2.10 | 13.86 | 16.32 | 2.05 | 2.95 | 6.35 | 0.30 |
| | Tango 2(Majumder et al., 2024) | T2A | 2.31 | 10.00 | 22.96 | 3.47 | **3.93** | 5.99 | 0.29 |
| | Stable Audio Open(Evans et al., 2024b) | T2A | 2.36 | 14.45 | 26.00 | 2.60 | 2.64 | **6.53** | **0.33** |
| | MAGNET-large(Ziv et al., 2024) | T2A | 2.03 | 8.53 | 22.17 | 2.74 | 3.65 | 5.25 | 0.26 |
| | MMAudio(Cheng et al., 2025) | T2A | 2.17 | 17.83 | 11.52 | 2.50 | 3.02 | 6.12 | 0.32 |
| | AudioX | T2A | **1.74** | **19.58** | **9.01** | **1.33** | 3.34 | 6.31 | **0.33** |
| | Seeing&Hearing(Xing et al., 2024) | V2A | 2.58 | 5.15 | 27.21 | 5.23 | 3.42 | 5.33 | **0.36** |
| | FoleyCrafter(Zhang et al., 2024) | V2A | 2.39 | 8.70 | 17.68 | 2.23 | 3.31 | 5.99 | 0.27 |
| | Diff-Foley(Luo et al., 2024) | V2A | 3.01 | 8.35 | 56.54 | 5.89 | 2.57 | 5.85 | 0.20 |
| | FRIEREN(Wang et al., 2024) | V2A | 2.58 | 6.91 | 50.88 | 3.13 | 2.98 | 6.06 | 0.20 |
| | MMAudio(Cheng et al., 2025) | V2A | **1.97** | **14.95** | 6.18 | 2.04 | 3.38 | 5.91 | 0.35 |
| | AudioX | V2A | 2.21 | 12.60 | **7.84** | **1.28** | **3.49** | **6.21** | 0.26 |
| | FoleyCrafter(Zhang et al., 2024) | TV2A | 1.94 | 11.32 | 19.16 | 2.13 | 3.38 | 6.06 | 0.26 |
| | MMAudio(Cheng et al., 2025) | TV2A | 1.51 | 17.79 | **6.60** | 2.20 | 3.31 | 5.99 | **0.33** |
| | AudioX | TV2A | **1.48** | **17.91** | 6.97 | **1.06** | **3.46** | **6.29** | 0.26 |
| AVVP | Seeing&Hearing(Xing et al., 2024) | V2A | 2.30 | 4.02 | 40.38 | 8.66 | 3.64 | 5.16 | **0.35** |
| | FoleyCrafter(Zhang et al., 2024) | V2A | 2.13 | 6.46 | 28.68 | 3.77 | 3.25 | 5.87 | 0.28 |
| | Diff-Foley(Luo et al., 2024) | V2A | 3.14 | 5.97 | 76.96 | 10.95 | 2.55 | 5.71 | 0.16 |
| | FRIEREN(Wang et al., 2024) | V2A | 2.73 | 4.71 | 66.46 | 6.49 | 3.08 | 5.88 | 0.17 |
| | MMAudio(Cheng et al., 2025) | V2A | **1.22** | 8.40 | 13.51 | 3.25 | 3.55 | 5.89 | 0.34 |
| | AudioX | V2A | 1.89 | **8.60** | 17.2 | **2.24** | **3.65** | **6.09** | 0.28 |
| | FoleyCrafter(Zhang et al., 2024) | TV2A | 1.81 | 6.22 | 26.76 | 2.85 | 3.62 | 5.60 | 0.27 |
| | MMAudio(Cheng et al., 2025) | TV2A | **1.74** | 9.52 | 14.18 | 2.74 | 3.64 | 5.81 | **0.34** |
| | AudioX | TV2A | 1.88 | **9.03** | 16.33 | **2.38** | **3.65** | **6.04** | 0.28 |
| MusicCaps | MusicGen(Copet et al., 2024) | T2M | 1.43 | 2.24 | 25.40 | 4.55 | 5.19 | 7.16 | 0.18 |
| | AudioLDM-L-Full(Liu et al., 2023) | T2M | 1.45 | 2.49 | 34.44 | 6.34 | 4.72 | 6.10 | 0.22 |
| | AudioLDM-2-Large(Liu et al., 2024a) | T2M | 1.26 | 2.84 | 15.61 | 2.80 | 5.22 | 6.70 | 0.23 |
| | TangoMusic(Ghosal et al., 2023) | T2M | 1.13 | 2.86 | 15.00 | 1.88 | 5.57 | 7.06 | 0.23 |
| | Stable Audio Open(Evans et al., 2024b) | T2M | 1.51 | 2.94 | 36.33 | 3.23 | 3.91 | **7.18** | 0.23 |
| | MAGNET-large(Ziv et al., 2024) | T2M | 1.32 | 1.98 | 23.88 | 4.24 | **5.84** | 6.71 | 0.19 |
| | AudioX | T2M | **0.96** | **3.55** | **9.76** | **1.53** | 5.21 | 6.70 | **0.24** |
| V2M-bench | MusicGen(Copet et al., 2024) | T2M | 0.76 | 1.31 | 40.59 | 3.25 | 5.57 | 7.43 | 0.14 |
| | AudioLDM-L-Full(Liu et al., 2023) | T2M | 0.72 | 1.37 | 36.63 | 2.97 | 5.08 | 7.01 | 0.16 |
| | AudioLDM-2-Large(Liu et al., 2024a) | T2M | 0.62 | 1.46 | 25.80 | **1.63** | 5.57 | 6.90 | 0.14 |
| | TangoMusic(Ghosal et al., 2023) | T2M | 0.72 | 1.46 | 38.19 | 2.43 | 5.78 | 7.46 | 0.14 |
| | Stable Audio Open(Evans et al., 2024b) | T2M | 0.72 | 1.34 | 42.02 | 2.72 | 4.36 | **7.72** | **0.17** |
| | MAGNET-large(Ziv et al., 2024) | T2M | 0.60 | 1.26 | 34.24 | 3.15 | **5.89** | 7.04 | **0.17** |
| | AudioX | T2M | **0.47** | **1.50** | 19.62 | 1.68 | 5.91 | 7.12 | 0.14 |
| | Video2Music(Kang et al., 2024) | V2M | 1.78 | 1.01 | 144.88 | 18.72 | 3.34 | 8.14 | 0.14 |
| | MuMu-LLaMA(Liu et al., 2024c) | V2M | 1.00 | 1.25 | 52.25 | 5.10 | 5.60 | 7.97 | 0.18 |
| | CMT(Di et al., 2021) | V2M | 1.22 | 1.24 | 85.70 | 8.64 | 4.98 | **8.20** | 0.12 |
| | VidMuse(Tian et al., 2025) | V2M | 0.73 | 1.32 | 29.95 | 2.46 | **5.88** | 6.89 | 0.20 |
| | AudioX | V2M | **0.70** | **1.37** | **24.01** | **1.67** | 5.24 | 7.04 | **0.23** |
| | AudioX | TV2M | 0.45 | 1.52 | 18.64 | 1.44 | 5.42 | 7.24 | 0.22 |

T2A-bench (detailed in Appendix A.3), we measure category, count, ordering, and timestamp accuracy (Cat-acc, Cnt-acc, Ord-acc, TS-acc). On AudioTime (Xie et al., 2025), we use its established metrics for Ordering, Duration, Frequency, and Timestamp.

**Subjective Evaluation.** We conducted a formal user study with 10 professional audio experts to evaluate the subjective quality of our generated samples against baselines. The study followed the established methodologies of prior work (Kreuk et al., 2022; Liu et al., 2023), where experts rated anonymized samples from 1 to 100 on Overall Quality (OVL) and Relevance (REL) to the prompt.

## 5.3 MAIN RESULTS

This work introduces a unified model capable of generating audio and music from flexible combinations of video, text, and audio inputs. Through extensive experimentation, we benchmark our model against SOTA specialist models across all supported tasks. Results demonstrate that our single model consistently achieves SOTA or highly competitive performance on the majority of metrics.

**Audio generation.** Results of our audio generation are in Table 1, which includes the outcomes of generating audio or music from any combination of video and text modalities. The upper part of the table presents the audio generation tasks, while the lower part displays the music generation tasks.

For text-to-audio generation, we evaluate on the AudioCaps (Kim et al., 2019) and VGGSound (Chen et al., 2020) datasets. On AudioCaps, our model achieves SOTA performance, while on VGGSound, the advantage is even more pronounced. This demonstrates that our model is a powerful text-to-audio generator. Furthermore, both our model and baseline results on VGGSound confirm the effectiveness of our curated caption data. For video-to-audio generation, we experiment on VGGSound and AVVP (Tian et al., 2020), AVVP is an out-of-domain test dataset for all methods. Our model achieves results comparable to SOTA on both VGGSound and AVVP, proving that it is not only a strong video-to-audio generator but also exhibits excellent generalization on out-of-domain datasets. For audio generation conditioned on both text and video, we benchmark against the strong baselines FoleyCrafter (Zhang et al., 2024) and MMAudio (Cheng et al., 2025), achieving results that are comparable to them. We find that when both text and video inputs are provided, the model effectively integrates the information from both modalities to generate better results.

The bottom part of Table 1 shows the results of music generation tasks. On the V2M dataset (Tian et al., 2025), we evaluate text-to-music, video-to-music, and video-and-text-to-music. The text-to-music task is additionally evaluated on the MusicCaps (Copet et al., 2024) dataset. Our model achieves SOTA performance across these tasks, demonstrating its effectiveness in generating high-quality music conditioned on diverse inputs.

Table 2: Evaluation of instruction-following T2A ability on the T2A-bench and AudioTime.

| Method | T2A-bench | | | | AudioTime | | | |
|---|---|---|---|---|---|---|---|---|
| | Cat-acc ↑ | Cnt-acc ↑ | Ord-acc ↑ | TS-acc ↑ | Ordering ↓ | Duration ↓ | Frequency ↓ | Timestamp↑ |
| AudioGen | 24.40 | 5.40 | 6.00 | 18.40 | 0.91 | 3.73 | 1.58 | 0.54 |
| AudioLDM | 18.60 | 4.00 | 3.40 | 11.60 | 0.97 | 3.41 | 1.54 | 0.41 |
| AudioLDM-2 | 20.10 | 7.40 | 1.20 | 13.40 | 0.96 | 3.40 | 1.64 | 0.54 |
| Tango 2 | 25.20 | 4.60 | 10.20 | 18.80 | 0.86 | 3.70 | 1.52 | 0.61 |
| Make-An-Audio2 | 32.40 | 4.00 | 19.80 | 18.80 | 0.76 | 3.40 | 1.42 | 0.56 |
| Stable Audio Open | 31.20 | 9.80 | 6.00 | 21.80 | 0.98 | 3.07 | 1.46 | 0.53 |
| MMAudio | 26.60 | 4.80 | 2.40 | 21.40 | 0.98 | 3.33 | 1.54 | 0.50 |
| AudioX | **34.20** | **12.40** | **23.60** | **28.20** | **0.34** | **1.30** | **0.74** | **0.81** |

**Instruction-following text-to-audio generation.** As shown in Figure 1 and Table 2, AudioX substantially outperforms all baselines in tasks requiring fine-grained control. On our T2A-bench, AudioX demonstrates a commanding lead across all dimensions, from category generation to count and temporal control. For instance, it surpasses the temporally-enhanced Make-An-Audio2 baseline in Ord-acc. This advantage is reaffirmed on the AudioTime benchmark. We also note that the performance trends between AudioTime's Ordering metric and our Ord-acc are consistent across all models, which helps validate the design of our benchmark for evaluating temporal adherence. Furthermore, an insight from these comparisons is that high audio fidelity does not necessarily correlate with instruction-following prowess. For instance, Tango2, despite its high-quality synthesis, delivers only moderate performance on these control-focused metrics. Collectively, these results underscore our model's superior fine-grained control, setting a new standard for controllable T2A generation.

**User study.** We conducted a user study to evaluate the quality of the generated audio and music. We randomly selected 25 samples for each audio generation task, including T2A, T2M, V2A, and V2M. 10 audio experts are asked to rate the quality of the generated audio and music. The results are shown in Fig. A.2 in the Appendix. The evaluation shows that our model achieves subjective SOTA performance in terms of OVL and REL scores in most tasks, indicating high user satisfaction.

To further demonstrate the versatility of our model, we present results for additional tasks, including audio inpainting, music completion, and image-to-audio generation, in Appendix A.4.1. The results further underscore our model's strong performance and broad applicability across a variety of audio generation tasks.

## 5.4 ABLATION STUDY

In this section, we conduct a series of ablation studies to investigate the contribution of our key design choices. We systematically validate the efficacy of our data curation strategy and the ar-

chitectural integrity of the proposed MAF module. An additional ablation study on the impact of different conditioning modalities is detailed in Appendix A.4.2.

**Efficacy of data curation strategy.**

To verify the impact of our data curation strategy, we evaluate models trained on different textual supervision sources (Table 3): 1) `Labels`: using raw class labels from the source datasets; 2) `AudioSetCaps`: using captions from a recent concurrent dataset (Bai et al., 2025); 3) `QwenCap`: using captions generated directly by Qwen2-Audio; 4) `GeminiCap`: using only the initial annotations generated by Gemini 2.5 Pro; and 5) `GeminiCap-aug`: our full pipeline. The results show that `GeminiCap-aug` outperforms all baselines, including the external AudioSetCaps dataset and the single-stage generation methods. It not only achieves the best scores on general-purpose tasks (T2A, V2A, TV2A) but also enhances the model's instruction-following capabilities. Collectively, these results validate the superior quality of our constructed dataset and the effectiveness of the proposed two-stage curation pipeline. Notably, we observe that the benefits of high-quality textual supervision are not limited to text-to-audio generation. The marked improvement in the V2A task provides strong empirical evidence of a ***cross-modal regularization effect***. This insight leads to a crucial conclusion for future work: high-quality textual data should be viewed not only as an input, but also as an effective strategy for building more capable and robust multimodal models.

Table 3: **Ablation study on data curation strategies.** We compare our model's performance when trained with captions from different sources. The results show a clear trend of improvement with higher-quality data. Our full pipeline (`GeminiCap-aug`) not only achieves the best performance on all general tasks (T2A, V2A, TV2A) but is also essential for enabling fine-grained control.

| Caption Method | Instruction-following T2A | | | T2A | | V2A | | TV2A | |
|---|---|---|---|---|---|---|---|---|---|
| | Cat-acc ↑ | Cnt-acc ↑ | Ord-acc ↑ | IS ↑ | FAD ↓ | IS ↑ | FAD ↓ | IS ↑ | FAD ↓ |
| Labels | 17.35 | 2.80 | 4.60 | 7.59 | 6.02 | 10.46 | 1.81 | 10.62 | 3.41 |
| AudioSetCaps | 27.85 | 6.40 | 4.80 | 10.08 | 3.19 | 11.35 | 1.33 | 12.39 | 1.56 |
| QwenCap | 24.60 | 6.40 | 6.20 | 9.74 | 4.40 | 10.57 | 1.67 | 11.79 | 1.95 |
| GeminiCap | 28.05 | 9.60 | 7.60 | 10.81 | 3.02 | 11.48 | 1.31 | 12.78 | 1.70 |
| GeminiCap-aug | **28.91** | **10.20** | **7.80** | **10.93** | **2.91** | **11.69** | **1.15** | **12.90** | **1.48** |

Table 4: **Ablation study of the MAF architecture components.** We evaluate the contribution of the Gate and Query mechanisms by removing them individually. The results show that the `Full MAF`, which includes both components, achieves the best performance across most metrics. This confirms that our complete design is essential for effective multimodal fusion.

| Components | Gate | Query | KL ↓ | IS ↑ | FD ↓ | FAD ↓ | Duration ↓ | Frequency ↓ | Ordering ↓ |
|---|---|---|---|---|---|---|---|---|---|
| w/o MAF | × | × | 1.83 | 10.70 | 11.60 | 2.67 | 3.022 | 1.359 | 0.912 |
| w/o Gate | × | ✓ | 1.69 | 11.66 | 9.72 | 2.00 | 2.945 | 1.348 | **0.876** |
| w/o Query | ✓ | × | 1.71 | 11.72 | 9.65 | 2.08 | 2.841 | 1.328 | 0.912 |
| Full MAF | ✓ | ✓ | **1.68** | **11.84** | **9.64** | **1.98** | **2.827** | **1.302** | 0.888 |

**Architectural ablation of the MAF module.** We conduct an architectural ablation of the MAF module to validate its design (Table 4). The results confirm that each component is integral, with the most severe performance deterioration observed when the MAF module is omitted entirely. Removing the Gate mechanism or the Query-based attention individually also results in a performance decline, confirming their respective contributions. This analysis validates our design choices, underscoring that the complete MAF architecture is critical for optimal multimodal fusion, thereby enhancing cross-modal alignment and improving generation quality.

## 5.5 DISCUSSION

Our extensive experiments provide a multi-faceted validation of AudioX, consistently demonstrating state-of-the-art performance from broad audio generation to a commanding lead in fine-grained instruction-following. Our ablation studies confirm that this success is directly attributable to two core principles: a data curation strategy that provides a rich semantic foundation via a powerful cross-modal regularization effect, and an MAF architecture essential for translating these signals into precisely controlled outputs. The synergy between this data-centric foundation and purpose-

built architecture culminates in our model's SOTA performance on challenging instruction-following benchmarks, validating our approach for unifying generative versatility with fine-grained control.

# 6 CONCLUSION

In this work, we present AudioX, a unified framework that transcends the modality and domain constraints prevalent in prior specialist models for audio generation. By leveraging a DiT backbone and our designed MAF module, our model effectively unifies diverse inputs like text, video, and audio to produce high-quality outputs. The training of our model is supported by IF-caps, our large-scale, fine-grained dataset, which provides a robust foundation for unified training and evaluation. Notably, our training methodology induces an effective cross-modal regularization effect, enhancing the model's internal representations. Extensive experiments demonstrate that our single, unified model not only matches or outperforms specialist models but also unlocks superior instruction-following capabilities, showcasing its command of both generative versatility and fine-grained control.

# 7 ACKNOWLEDGMENT

The research was supported by the Early Career Scheme (ECS-Hong Kong University of Science and Technology 22201322), the Theme-based Research Scheme (T45-205/21-N), as well as the Generative AI Research and Development Center from InnoHK. We thank Fortuna Peng for helpful discussions and Jingyi Li for assistance with the figures.

## ETHICS STATEMENT

The authors have adhered to the ICLR Code of Ethics. Our work is intended to foster positive and creative applications in media, and we have carefully considered the ethical implications of our research. The IF-caps dataset is curated exclusively from publicly available sources, and its release will fully comply with the licenses of the original material. Our human evaluation involved compensated professional experts who provided informed consent. No conflicts of interest or sponsorship bias exist in this work, and all authors adhere to research integrity practices, including transparent documentation of data sources, collection procedures, and evaluation protocols. By committing to open-sourcing our code, model, and dataset, we aim to ensure transparency and support the community in the responsible advancement of generative audio technology.

## REPRODUCIBILITY STATEMENT

We are committed to ensuring the reproducibility of our work and have provided comprehensive details of our methodology and experiments. The complete architecture of our AudioX framework, including the Multimodal Adaptive Fusion module, is detailed in Section 4.1 and visualized in Figure 4. All implementation details, including optimizer settings, learning rates, batch sizes, and computational resources, are provided in Section 5.1. Our data curation pipeline for constructing the IF-caps dataset is described in Section 3, with further details on the annotation schema, data statistics, and augmentation samples available in Appendix A.1.1. Our full evaluation protocol, including precise definitions for all objective and subjective metrics and the design of our new benchmark, T2A-bench, is detailed in Section 5.2, Appendix A.2, and Appendix A.3. To further aid replication and future research, we will open-source our code, pretrained model checkpoints, and the complete IF-caps dataset upon publication.

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

# A APPENDIX

## USE OF LARGE LANGUAGE MODELS

Large Language Models (LLMs) are utilized in two parts in this work. As components of our methodology, Gemini 2.5 Pro is employed for high-quality initial data annotation, benchmark generation, and as an automated evaluator, while Qwen2-Audio performs large-scale data augmentation. Additionally, an LLM assistant (Google's Gemini) is used as a writing tool to improve the clarity, grammar, and vocabulary of the manuscript. All scientific ideation, experimental design, and analysis are conceived and performed exclusively by the human authors.

## APPENDIX OVERVIEW

This appendix supplements the main paper with expanded details on our datasets, evaluation methodologies, and a broader range of experimental results. We begin by detailing our data and evaluation frameworks: Section A.1 delves into the specifics of our datasets and the annotation process, Section A.2 introduces evaluation metrics, while Section A.3 introduces T2A-bench, our benchmark for instruction-following, along with its automated evaluation pipeline. Subsequently, we present an expanded set of results. Section A.4 provides further quantitative comparisons, and finally, Section A.5 showcases a comprehensive gallery of qualitative examples and analyses.

## A.1 DATASETS

### A.1.1 TRAINING AND TEST DATASETS

Table A.1 provides an overview of all datasets used in this work. Table A.2 outlines the new captions we annotated for training and testing our unified model. We will open-source these caption datasets to facilitate further research.

### A.1.2 FURTHER DETAILS ON THE IF-CAPS DATASET

As described in the main text, the IF-caps dataset is generated via a multi-step pipeline designed to produce rich, structured annotations for existing video-audio clips. This section provides a detailed breakdown of our annotation schema and showcases representative samples.

**Annotation Schema.** Each sample in IF-caps is accompanied by a comprehensive set of annotations designed to provide multi-faceted supervision for training. The key fields are as follows:

- **caption**: A holistic, high-level natural language description of the audio content, summarizing the main events and their context.

- **category**: A structured dictionary that provides sound event classification and, where applicable, the discrete count of each event. For continuous or unquantifiable sounds (e.g., background noise, speech), the count is marked as null.

- **SED (Sound Event Detection)**: A list providing fine-grained temporal localization. Each entry in the list maps a precise timestamp (e.g., "00:02-00:06") to a description of the sound event occurring within that specific time frame.

- **time_relation**: A field describing the temporal relationship between distinct sound events. This can specify a sequential order (e.g., "Event A, Event B") or more complex relationships like "interleave" for overlapping sounds.

This structured format allows our model to learn not just what sounds are present, but also how many, when, and in what order, which is critical for developing advanced instruction-following capabilities.

**Annotation Samples.** Below are two examples from IF-caps that illustrate the richness and detail of our annotation schema. The first example demonstrates a complex scene with overlapping, continuous, and countable events. The second example shows a clear sequence of discrete events.

Table A.1: Comprehensive overview of training and test datasets, detailing the number of clips (# Clips), average duration per clip (Dur./Clip in seconds), and total duration (Dur. in hours) for each task and split. T2A: Text-to-Audio, V2A: Video-to-Audio, TV2A: Text-and-Video-to-Audio, T2M: Text-to-Music, V2M: Video-to-Music, TV2M: Text-and-Video-to-Music.

| Split | Task | Dataset | # Clips | Dur./Clip (s) | Dur. (h) |
|---|---|---|---|---|---|
| Train | T2A | AudioCaps | 45.0k | 10 | 125.1 |
| | | WavCaps | 108.3k | 10 | 300.8 |
| | | IF-caps | 1268k | 10 | 3524.4 |
| | | AudioTime | 20k | 10 | 355.5 |
| | V2A | VGGSound | 176.9k | 10 | 491.4 |
| | | AudioSet Strong | 67.3k | 10 | 187.14 |
| | | Greatest Hits | 1.0k | 10 | 2.71 |
| | TV2A | IF-caps | 1268k | 10 | 3524.4 |
| | | Greatest Hits | 1.0k | 10 | 2.71 |
| | T2M | Private | 175.2k | 240 | 11679.3 |
| | | V2M | 5685.7k | 10 | 15793.58 |
| | | MUCaps | 22.0k | 208 | 1273.6 |
| | V2M | V2M | 5685.7k | 10 | 15793.58 |
| | TV2M | V2M | 5685.7k | 10 | 15793.58 |
| | Audio Inpainting | All audio data | 398.5k | 10 | 1107.15 |
| | Music Completion | All music data | 5882.9k | 17.6 | 28746.48 |
| Test | T2A | AudioCaps | 4,875 | 10 | 13.54 |
| | | VGGSound | 14,931 | 10 | 41.475 |
| | | T2A-bench | 2000 | 10 | 5.55 |
| | | AudioTime | 2000 | 10 | 5.55 |
| | V2A | VGGSound | 14,931 | 10 | 41.475 |
| | | AVVP | 1,120 | 10 | 3.11 |
| | TV2A | VGGSound | 14,931 | 10 | 41.475 |
| | T2M | MusicCaps | 5,526 | 10 | 15.35 |
| | | V2M | 3105 | 10 | 9.01 |
| | V2M | V2M | 300 | 108 | 9.01 |
| | TV2M | V2M | 300 | 108 | 9.01 |
| | Audio Inpainting | AudioCaps | 4,875 | 10 | 13.54 |
| | | AVVP | 1,120 | 10 | 3.11 |
| | Music Completion | V2M | 300 | 108 | 9.01 |

Table A.2: Overview of our labeled captions, detailing the number of clips, average duration per clip, and total duration for each source dataset.

| Source Dataset | Data Type | # Clips | Dur./Clip (s) | Dur. (h) |
|---|---|---|---|---|
| VGGSound | Audio | 191.8K | 10 | 532.81 |
| AudioSet Strong | Audio | 67.3K | 10 | 187.14 |
| AVVP Test Split | Audio | 1.1K | 10 | 3.11 |
| Greatest Hits | Audio | 1.0K | 10 | 2.71 |
| V2M | Music | 5.7M | 10 | 15793.58 |

```
Example 1:

{
"caption": "A woman is speaking continuously, while a dog yips
        twice in the background.",
"category": {
```

```
            "Female speech": null,
            "Yip": 2,
            "Background noise": null
            },
    "SED": [
        {"00:00-00:09": "A woman is speaking throughout the audio,
            accompanied by faint background noise."},
        {"00:00-00:01": "A dog lets out a yip in the background."},
        {"00:08-00:09": "A dog yips again in the background."}
        ],
    "time_relation": "interleave",
    "audio_id": "TATdZPmzMU8_90000"
    }
```

```
Example 2:

{
"caption": "The audio features the mechanical sound of a firearm
    being handled, immediately followed by two separate bursts of
    machine gun fire.",
"category": {
        "Machine gun": 2,
        "Generic impact sounds": 1
        },
"SED": [
        {"00:00-00:01": "The mechanical sound of a firearm being
            handled, possibly being cocked or loaded."},
        {"00:01-00:05": "A sustained burst of automatic gunfire from
            a machine gun."},
        {"00:06-00:08": "A second, shorter burst of machine gun fire
            ."}
        ],
"time_relation": "Generic impact sounds, Machine gun",
"audio_id": "c9OnubhhvZY_0"
}
```

**Data Augmentation Process.** As mentioned in the main text, a key step in our pipeline is to leverage a cost-effective model (Qwen2-Audio) to augment the initial, high-quality annotations generated by Gemini 2.5 Pro. The goal is to increase the linguistic and structural diversity of our dataset. By generating multiple, semantically equivalent but stylistically different captions for the same audio clip, we train our model to be robust to variations in user prompts and to develop a more generalized understanding of the relationship between language and sound. The augmentation process is guided by the structured fields of the original annotation. The model is prompted to generate new captions from different perspectives: rephrasing the original description, or generating new descriptions based purely on the category and count, the SED timestamps, or the time_relation fields. Below, we use the second example from the previous section to illustrate this structured augmentation process.

```
Original Audio Annotation (Generated by Gemini 2.5 Pro)

{"caption": "The audio features the mechanical sound of a firearm
    being handled, immediately followed by two separate bursts of
    machine gun fire.", "category": {"Machine gun": 2, "Generic
    impact sounds": 1}, "SED": [...], "time_relation": "Generic
    impact sounds, Machine gun", "audio_id": "c9OnubhhvZY_0"}
```

This single, rich annotation serves as the seed for generating a variety of new training captions, each emphasizing a different aspect of the audio content.

---

**Augmented Audio Captions (Generated by Qwen2-Audio)**

**1. Caption Rephrasing (Linguistic Diversity)**

*"A gun is cocked, followed by two bursts of machine gun fire."*
*"After the sharp, metallic sound of a firearm mechanism, two rapid-fire bursts from a machine gun are heard."*

**2. Augmentation from Category and Count**

*"The audio contains two sounds of a machine gun and one generic impact sound."*

**3. Augmentation from SED**

*"The sound of a firearm being handled is audible for the first second, followed by a burst of machine gun fire from 1 to 5 seconds and a second burst from 6 to 8 seconds."*

**4. Augmentation from Time Relation**

*"In this audio, the sound of a generic impact occurs first, followed by two distinct machine gun sounds."*

---

This structured augmentation strategy ensures our model is exposed to a wide variety of textual descriptions, learning to associate not only high-level captions but also explicit instructions about count, timing, and order with the corresponding audio features. Similarly, for music data, this process generates varied descriptions of genre, mood, instrumentation, and tempo, teaching the model to comprehend both high-level artistic direction and specific musical components.

---

**Original Music Annotation (Generated by Gemini 2.5 Pro)**

```
{"caption": "A heartwarming acoustic track featuring a blend of
    softly strummed guitar and a simple, melodic piano line,
    creating a gentle and uplifting atmosphere.", "genre": "Acoustic
     Pop, Instrumental", "mood": "Heartwarming, Gentle, Uplifting",
    "instrument": ["Acoustic Guitar", "Piano"], "tempo": "Slow to
    Moderate"}
```

---

This structured music annotation is then used to generate diverse new training captions, each focusing on a different attribute:

---

**Augmented Music Captions (Generated by Qwen2-Audio)**

**1. Caption Rephrasing**

*"A gentle instrumental piece with the interwoven sounds of an acoustic guitar and piano."*
*"Soft guitar strumming and a simple piano melody combine to create an uplifting acoustic pop track."*

**2. Augmentation from Genre**

*"An instrumental acoustic pop track featuring piano and guitar."*

**3. Augmentation from Mood**

*"A heartwarming, gentle, and uplifting piece of music featuring acoustic guitar and piano."*

**4. Augmentation from Tempo**

*"A slow to moderate tempo instrumental track with piano and acoustic guitar."*

---

## A.2 Details of evaluation metrics

**Fréchet Audio Distance (FAD).** To evaluate the perceptual quality of the generated audio, we employ FAD, a reference-free metric analogous to the FID (Heusel et al., 2017) score used in image generation. The metric functions by comparing the statistical distance between embedding distributions of generated audio and real-world audio. A smaller distance suggests the generated audio is of higher acoustic quality. For our calculations, we utilize the VGGish (Hershey et al., 2017) feature extractor.

**Fréchet Distance (FD).** While similar in principle to FAD, FD serves as a distinct measure of audio similarity by employing a different feature extractor. We use an FD variant based on PANNs (Kong et al., 2020) embeddings. Given that PANNs models are pretrained on the extensive AudioSet (Gemmeke et al., 2017), this metric is considered to be highly robust for evaluating audio fidelity.

**Kullback-Leibler Divergence (KL).** The KL divergence is used to approximate the acoustic similarity between generated and reference audio samples. This is achieved by measuring the divergence between the multi-label class prediction distributions produced by a PANNs model for both sets of samples.

**Inception Score (IS).** The IS is a widely used metric to evaluate the performance of generative models. Besides assessing the diversity of the generated samples, IS also evaluates their quality, measuring the clarity and recognizability of individual audio events (Donahue et al., 2018; Majumder et al., 2024; Liu et al., 2023). Given its ability to provide a single, holistic score reflecting both of these aspects without needing a reference prompt, we selected IS as the unified metric for the comprehensive performance comparison in our teaser Fig. 1. This allows for a fair and consistent visualization of our model's capabilities across the wide array of supported tasks.

**ImageBind Score (Girdhar et al., 2023).** We assess the semantic alignment between generated audio and conditioning videos using the ImageBind Score. This score is calculated as the cosine similarity between the audio and video embeddings from the respective branches of the ImageBind model.

**CLAP Score.** The Contrastive Language-Audio Pretraining (CLAP) model (Elizalde et al., 2023) learns a joint embedding space where audio clips and their corresponding text descriptions are aligned. We use the CLAP Score to evaluate the semantic alignment between generated audio and a text prompt, calculated as the cosine similarity between their respective embeddings from the pretrained CLAP encoders (Wu et al., 2023b). A higher score indicates better alignment.

**Production Complexity (PC) and Production Quality (PQ).** These metrics are derived from the Meta Audiobox Aesthetics framework (Tjandra et al., 2025). PQ focuses on the objective, technical aspects of an audio recording, such as its clarity, fidelity, dynamics, and frequency balance. In contrast, PC evaluates the complexity of an audio scene by measuring the number of distinct audio components present, such as multiple instruments or the co-occurrence of speech, music, and sound effects. Both are designed as no-reference metrics, allowing for the assessment of individual audio clips without needing a ground-truth comparison sample.

**Ordering, Duration, Frequency, and Timestamp.** These metrics are components of the STEAM evaluation framework, proposed in the AudioTime (Xie et al., 2025) to assess the temporal controllability of audio generation models. Ordering is an error rate that measures whether sound events are generated in the specified sequence. Duration and Frequency are calculated as the L1 error between the specified and detected event durations and occurrence counts, respectively. Timestamp evaluates the precise timing of events (onset and offset) using the F1-score, a common metric in sound event detection.

**Category, Count, Ordering, and Timestamp accuracy.** See A.3.

**Overall Quality (OVL) and Relevance (REL).** For our subjective evaluation, 10 professional audio experts rated each generated sample on a scale of 1 to 100 on two standard criteria. OVL assesses the intrinsic perceptual fidelity of the audio itself—focusing on aspects like clarity and freedom from artifacts—independent of the prompt. In parallel, REL measures the semantic alignment between the audio and its conditioning input, evaluating how accurately the content reflects the instructions from the provided text or video. This evaluation protocol follows the established methodologies of

prior work (Kreuk et al., 2022; Liu et al., 2023). Example of the questionnaire interface is shown in Table A.3.

Table A.3: Simplified example of the questionnaire for human evaluation, showcasing the four main task types. Experts provided scores for OVL and REL.

| File Name | Prompt (Text or Video) | OVL (1-100) | REL (1-100) |
|---|---|---|---|
| 9964.wav | A loud white noise and then some beeping. | 55 | 65 |
| 0928.wav | An uplifting folk-pop instrumental track. | 70 | 55 |
| 1441.wav | [Video of a person walking on dry leaves] | 80 | 70 |
| 1701.wav | [Video of a drone shot over a sunrise mountain] | 65 | 60 |
| ... | ... | ... | ... |

## A.3 BENCHMARK AND METRICS FOR INSTRUCTION-FOLLOWING IN T2A

To rigorously and scalably evaluate the instruction-following capabilities of Text-to-Audio generation models, we introduce a new benchmark, **T2A-bench**, and a corresponding automated evaluation pipeline. This framework is designed to dissect a model's ability to adhere to complex compositional instructions.

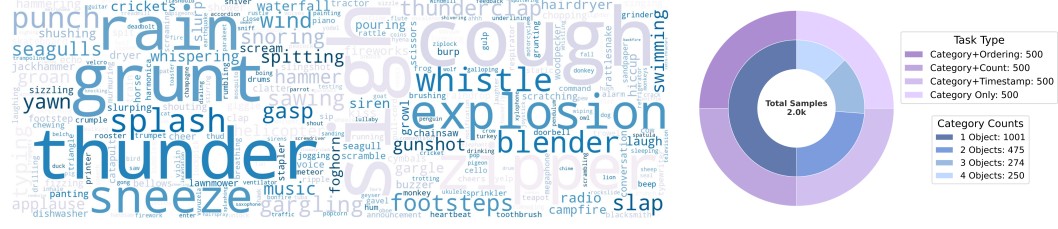

(a) Sound category word cloud.    (b) Task & Category distribution.

Figure A.1: The composition of the T2A-bench benchmark. (a) Word cloud of sound event categories. (b) Distribution of task types and category counts.

**T2A-bench Composition and Design.** T2A-bench is a prompt-based benchmark comprising 2k challenging, natural language prompts generated by Gemini 2.5 Pro. It is structured to systematically probe four key dimensions of controllability. As illustrated in Figure A.1, our benchmark encompasses a diverse vocabulary of sound categories and a balanced task structure to enable a rigorous and comprehensive evaluation. The benchmark is divided into four task types, each containing 500 prompts:

- **Category-only**: Evaluates the generation of correct sound events. Prompts contain between one and five distinct sound categories (100 prompts for each count).

- **Category+Count**: Assesses the ability to generate a precise number of sound events. To avoid ambiguity, prompts in this category feature only a single sound type, with the required count ranging from one to five (100 prompts for each count).

- **Category+Ordering**: Measures adherence to temporal sequence. Prompts specify an order for either two or three distinct sound categories.

- **Category+Timestamp**: Tests temporal localization. To ensure clarity, prompts specify a start and end time for a single sound category.

Below are representative examples for each task type, including the prompt and its corresponding structured metadata.

```
T2A-bench Examples

{
"id": "T2A_01565",
"type": "category-only",
"prompt": "A violent storm at sea, with a loud clap of thunder and a
    huge wave crashing over the deck.",
"category": "thunder, wave crash"
}
{
"id": "T2A_00031",
"type": "category+count",
"prompt": "A single, loud bark from a dog in the distance.",
"category": "dog bark",
"count": {"dog bark": 1}
}
{
"id": "T2A_00575",
"type": "category+ordering",
"prompt": "The sound of a person gargling, followed by the splash of
    water in the sink.",
"category": "gargle, water splash",
"time_relation": "gargle, water splash"
}
{
"id": "T2A_01105",
"type": "category+timestamp",
"prompt": "The sound of a crowd cheering is present from 2.0 seconds
    to 6.0 seconds.",
"category": "crowd cheering",
"timestamp": {"crowd cheering": {"start": 2.0, "end": 6.0}}
}
```

**Evaluation Metrics.** Corresponding to the benchmark's structure, we define four strict, accuracy-based metrics: Category Accuracy (Cat-acc), Count Accuracy (Cnt-acc), Ordering Accuracy (Ord-acc), and Timestamp Accuracy (TS-acc). The final score for each metric is the percentage of "correct" judgments.

- **Cat-acc**: A judgment is "correct" only if *all* sound categories specified in the prompt are detected in the generated audio. This is evaluated on all 2,000 samples.

- **Cnt-acc**: A judgment is "correct" only if the detected count for the specified category exactly matches the prompt's instruction.

- **Ord-acc**: A judgment is "correct" only if the detected temporal order of sound events exactly matches the specified sequence.

- **TS-acc**: A judgment is "correct" only if the detected event's start and end times fall within a 1-second tolerance window of the target times specified in the prompt.

**Automated Evaluation Pipeline.** To ensure objective and scalable evaluation while preventing information leakage, we designed a novel two-step pipeline that leverages the state-of-the-art audio understanding of a powerful Multimodal Large Model (MLLM), Gemini 2.5 Pro, as an automated judge.

- **Step 1: Blind Audio Annotation.** In the first step, the MLLM judge receives *only* the audio sample generated by the model under evaluation. It performs a blind, detailed analysis to produce a structured annotation of the audio's content. This annotation includes detected sound categories, their counts, temporal relationships, and precise sound event detection (SED) timestamps. For sounds where counting is ambiguous (e.g., continuous water flow) or ordering is not distinct, the corresponding fields are populated with null.

```
Example of Step 1 Output (Structured Annotation)

{
"caption": "The audio contains two distinct loud sounds.
    First, there is a deep, rolling thunderclap. After a
    brief pause, a powerful and sudden explosion is heard.",
"category": {"Thunder": 1, "Explosion": 1},
"SED": [
        {"00:01.734-00:03.514": "A deep, rolling thunderclap
            is heard."},
        {"00:08.241-00:09.511": "A loud and sudden explosion
            with a distinct boom."}],
"time_relation": "Thunder, Explosion",
}
```

- **Step 2: LLM-based Judgment.** In the second step, the MLLM judge is provided with the original prompt from T2A-bench and the structured annotation generated in Step 1. Acting like an examiner with an answer key, the MLLM compares the annotated audio content against the prompt's instructions. It then outputs a binary score (1 for correct, 0 for incorrect) for the relevant metric, along with a detailed textual analysis explaining its decision.

```
Example of Step 2 Output (Final Judgment)

{
"prompt": "A medieval battlefield, with the sound of a
    catapult launching a stone and the subsequent explosion."
"prediction": {"cat_acc": 0, "cnt_acc": null, "ord_acc": null
    , "ts_acc": null,
"analysis": "The audio contains a clear and prominent sound
    of thunder, which is audible from the beginning and
    culminates in a loud clap around 00:03. However, the
    required category 'wave crash' is missing. While there is
     a sound of water starting around 00:05, it is
    acoustically identifiable as heavy rain rather than a
    distinct, powerful wave crashing."}
}
```

In summary, our framework, combining T2A-bench, fine-grained metrics, and a robust two-step evaluation pipeline, provides a comprehensive and replicable methodology for quantifying the instruction-following capabilities of T2A models. We will open-source our proposed benchmark and evaluation pipeline to facilitate future research in this area.

## A.4   MORE RESULTS

### A.4.1   COMPARISON RESULTS

**Audio inpainting.** As shown in Table A.4, we conducted experiments on audio inpainting tasks, where our model outperformed the baselines (Liu et al., 2023; 2024a) on the AudioCaps (Kim et al., 2019) and AVVP (Tian et al., 2020) test datasets. Additionally, to explore audio inpainting with various input modalities, we performed experiments on unconditioned audio inpainting, as well as video-guided and text-and-video-guided audio inpainting tasks (on AVVP). The results indicate that both text and video can effectively guide the audio inpainting task, with text providing better guidance than video. When both text and video are conditioned, the model can integrate the two modalities to achieve superior results.

**Music Completion.** Music completion is a task where the model generates music based on a given music clip. We evaluate our model on the V2M-bench (Tian et al., 2025) dataset. The results are shown in Table A.5. We find that our model can generate music that extends the input music clip. As the number of input modalities increases, the model's performance improves, demonstrating its

Table A.4: **Inpainting Performance Comparison.** This table shows the performance comparison for audio inpainting on the AudioCaps and AVVP datasets. The values before and after the slash represent the IS and FAD metrics, respectively. A, V, and T represent Audio, Video, and Text conditions. The baseline methods are all under audio and text conditions.

| Method | Input | Dataset | |
|---|---|---|---|
| | | AudioCaps | AVVP |
| Unprocessed | - | 6.51/11.34 | 4.94/6.70 |
| AudioLDM-L-Full(Liu et al., 2024a) | A+T | 8.06/2.64 | 5.11/3.30 |
| AudioLDM-2-Full-Large(Liu et al., 2024a) | A+T | 4.24/10.17 | 3.99/11.58 |
| AudioX | A | 4.63/5.35 | 3.94/5.44 |
| AudioX | A+T | **9.84/2.25** | 6.12/2.05 |
| AudioX | A+V | N/A | 5.63/2.16 |
| AudioX | A+T+V | N/A | **6.25/1.99** |

strong inter-modal learning capability and ability to leverage multi-modal information to generate better music.

Table A.5: **Performance for our method under different conditions in the music completion task.** M, T, and V represent Music, Text, and Video, respectively.

| Input | KL ↓ | IS ↑ | FD ↓ | FAD ↓ |
|---|---|---|---|---|
| M | 0.96 | 1.21 | 52.77 | 5.76 |
| T+M | 0.51 | 1.49 | 21.42 | 2.14 |
| V+M | 0.70 | 1.37 | 24.28 | 2.29 |
| T+V+M | **0.46** | **1.52** | **18.69** | **1.67** |

**Image-to-audio generation.** To evaluate the model's capability in handling static visual inputs, we conduct a **zero-shot image-to-audio generation** experiment. Adopting the experimental protocol of Seeing&Hearing (Xing et al., 2024), we perform evaluations on 3k clips from the VGGSound test set, where keyframes were processed using AnimeGANv2 (Chen, 2022) to transfer them into "Paprika style" prior to generation. For comparison, we benchmark AudioX against Seeing&Hearing (Xing et al., 2024), Im2Wav (Sheffer & Adi, 2023), and also constructed a baseline by combining an image caption model (Bai et al., 2023) with a text-to-audio model (Majumder et al., 2024). The results are shown in Table A.6 in the Appendix. We find that our model demonstrates excellent performance in the image-to-audio generation task even without any specific training with image data.

Table A.6: **Comparison of Methods for the Image2Audio Task.**

| Method | KL ↓ | IS ↑ | FD ↓ | FAD ↓ | Align. ↑ |
|---|---|---|---|---|---|
| Caption2Audio | 2.76 | 7.48 | 32.97 | 5.54 | 0.21 |
| Im2Wav(Sheffer & Adi, 2023) | **2.61** | 7.06 | 19.63 | 7.58 | **0.41** |
| Seeing&Hearing(Xing et al., 2024) | 2.69 | 6.15 | 20.96 | 6.87 | 0.29 |
| AudioX | 2.90 | **13.48** | **16.42** | **2.71** | 0.23 |

**User study.** We conducted a user study to evaluate the quality of the generated audio and music. We randomly selected 25 samples for each audio generation task, including text-to-audio (T2A), text-to-music (T2M), video-to-audio (V2A), and video-to-music (V2M). 10 audio experts are asked to rate the quality of the generated audio and music. The results are shown in Fig. A.2. The evaluation shows that our model achieves subjective SOTA performance in terms of OVL and REL scores in most tasks, indicating high user satisfaction.

### A.4.2 ABLATION RESULTS

**Unified model performance.**

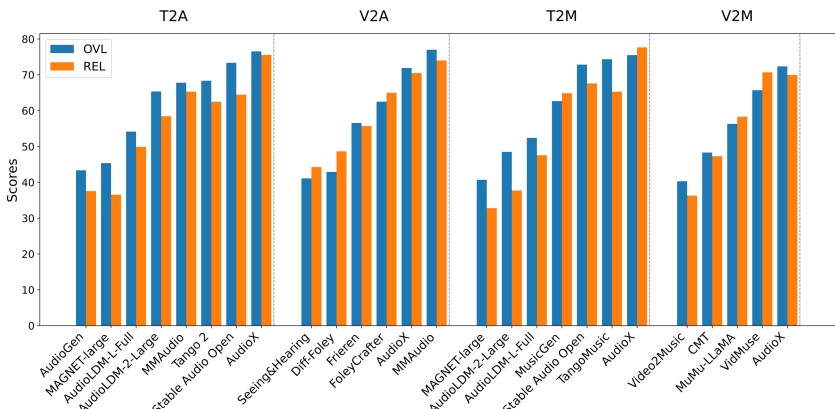

Figure A.2: User study results of generated audio and music. The values represent the average OVL and REL scores across Text-to-Audio (on AudioCaps), Text-to-Music (on MusicCaps), Video-to-Audio (on VGGSound), Video-to-Music (on V2M-bench).

We investigate our unified model's intra- and inter-modal performance in Fig. A.3. For the intra-modal study, we compare our single unified model against specialist models trained on individual tasks (T2A, V2A, and audio inpainting). The results show our unified model consistently outperforms these specialist models, demonstrating strong intra-modal capabilities. For the inter-modal study on music generation, we find that performance progressively improves as more conditioning modalities are added (e.g., from video-only to video+text). This confirms the model's robust ability to effectively integrate multiple modalities to enhance generation quality.

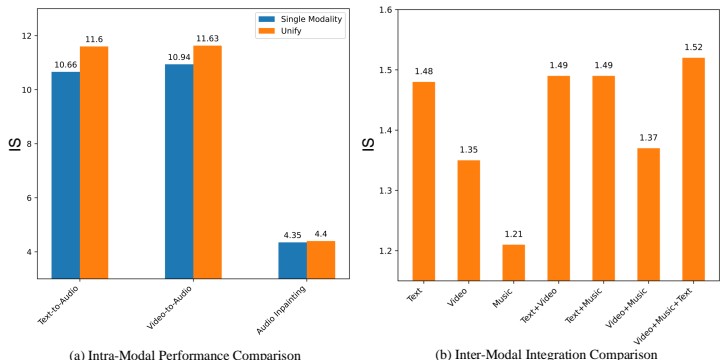

(a) Intra-Modal Performance Comparison

(b) Inter-Modal Integration Comparison

Figure A.3: Ablation study comparing intra-modal and inter-modal performance of the unified model. The left compares single-modality models on text-to-audio, video-to-audio, and audio inpainting tasks. The right shows the effect of adding modalities on music generation, with performance improvements noted for each added modality. Results are based on the Inception Score (IS) metric.

## A.5 QUALITATIVE RESULTS

Figures A.4 and A.5 present comprehensive qualitative results.

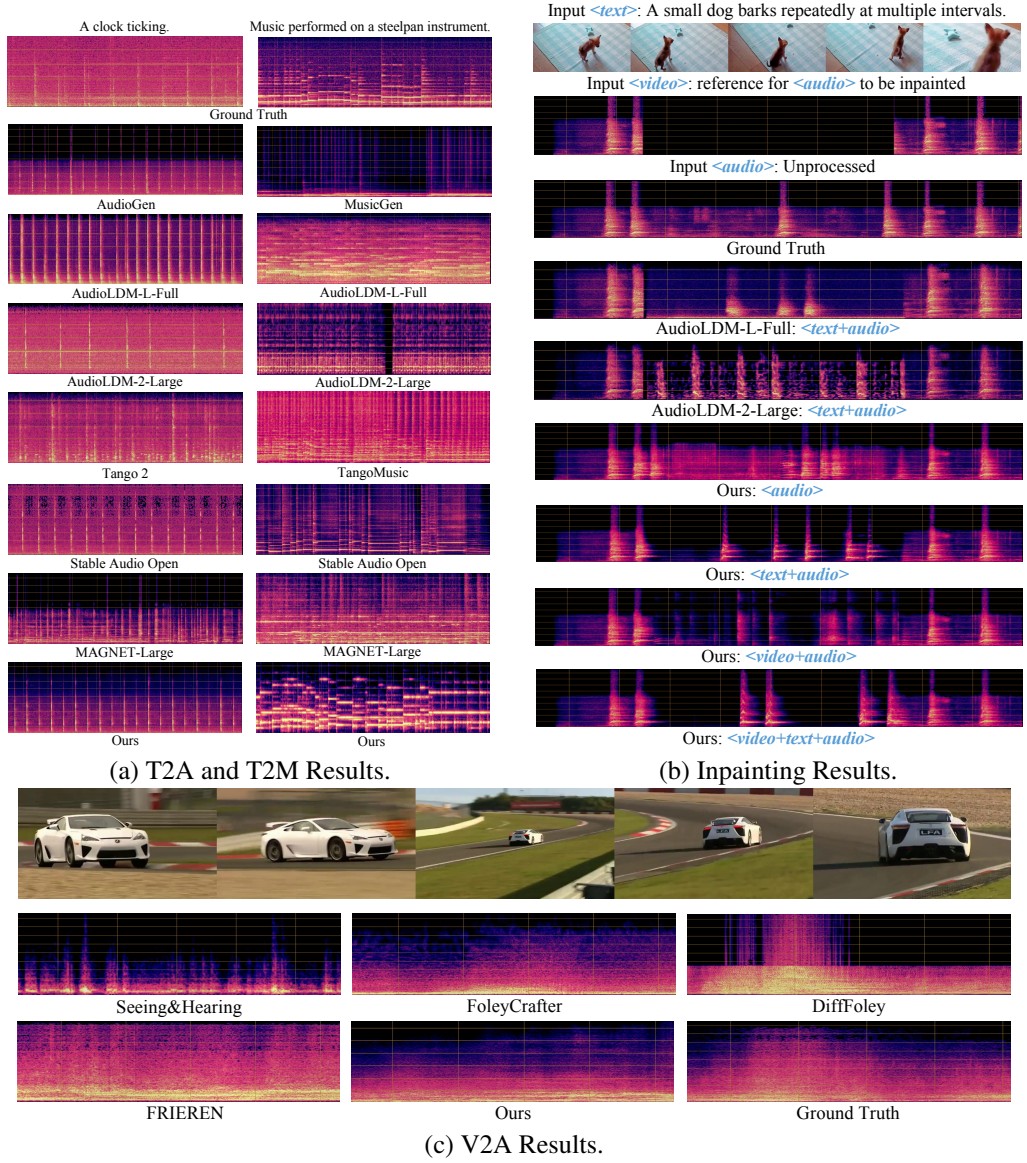

(a) T2A and T2M Results.

(b) Inpainting Results.

(c) V2A Results.

Figure A.4: Qualitative comparison across various tasks: (a) In Text-to-Audio (T2A) and Text-to-Music (T2M) tasks, our model uniquely excels by consistently generating the "ticking" sound of a clock and accurately following the prompt "Music performed on a steelpan instrument," outperforming baselines in both rhythmic precision and genre fidelity. (b) Audio inpainting results demonstrate our model's strong context-aware capabilities and its ability to effectively integrate different input modalities. (c) Video-to-Audio (V2A) results show our model's proficiency in capturing dynamic motion sounds, such as the immersive "drifting" of a car, providing a richer auditory experience compared to baselines.

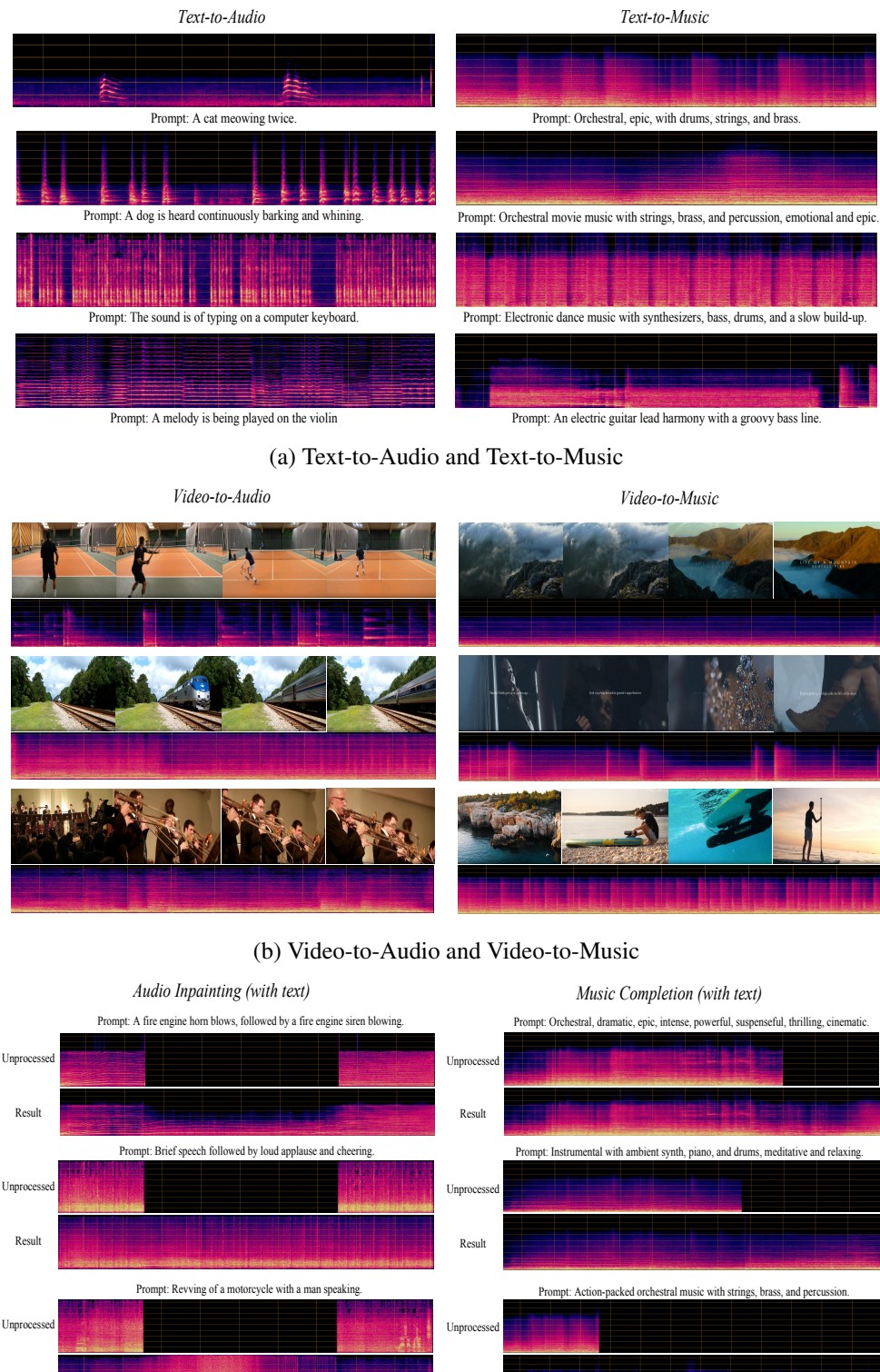

Figure A.5: Comprehensive qualitative analysis of our model's performance across various tasks: (a) Text-to-Audio and Text-to-Music synthesis, (b) Video-to-Audio and Video-to-Music generation, and (c) Audio Inpainting and Music Completion.

