# OpenReview forum: "AudioX: A Unified Framework for Anything-to-Audio Generation"
_ICLR.cc/2026/Conference — ICLR 2026 Poster_

### Official Review · Reviewer_CYKM · 2025-10-29

**Soundness:** 3
**Presentation:** 3
**Contribution:** 3
**Rating:** 8
**Confidence:** 5

**Summary:**

The paper proposes a unified DiT model for anything-to-audio generation conditioned on text / video / image / audio. The model adds a lightweight Multimodal Adaptive Fusion (MAF), per-modality gates and learnable query sets with cross/self-attention, to calibrate condition embeddings before a DiT generator. To train the model, the authors proposed a large-scale dataset, IF-caps, sourced from existing joint audio-video dataset, by using frontier audio-video LLMs for providing detailed captions for audio / music.

**Strengths:**

1. Unified model architecture: A single model handling text/video/audio conditions for both sound effects and music is non-trivial, and it seems like the results are promising.

2. Detailed evaluations and benchmarking: Evaluations cover many tasks (T2A/V2A/TV2A; T2M/V2M/TV2M; inpainting; completion; image->audio zero-shot), and the results look solid.

3. Novel evaluation prototype: beyond fidelity and diversity (FD / KL metrics), they propose to evaluate more fine-grained aspects,  temporal/count/order control (incl. AudioTime) and adds a new benchmark (T2A-bench).

4. Detailed user studies: the supplementary material provides detailed user studies and qualitative results over different aspects of the proposed method, covering a wide variety of tasks.

**Weaknesses:**

1. Lack of reference / comparisons against some important prior works in V2A / V2M, including but not limited to [1] - [5]. Except [3], all the rest are either open-sourced or providing samples for evaluation comparison. And also, [4] provides a new benchmark, which I think should be considered to evaluate.

2. Lack of important metrics regarding temporal alignment evaluation. The main table (Table 1) should also incorporate metrics that address temporal alignment, such as ones "Align Acc" proposed in [6] and "AlignSync" proposed in [7].




[1]. Tell what you hear from what you see-video to audio generation through text, NeurIPS 2024, Xiulong Liu, et.al.

[2]. From vision to audio and beyond: A unified model for audio-visual representation and generation. Kun Su, et.al.

[3]. V2meow: Meowing to the visual beat via video-to-music generation, AAAI 2024, Kun Su, et.al.

[4]. Kling-Foley: Multimodal Diffusion Transformer for High-Quality Video-to-Audio Generation.

[5]. ThinkSound: Chain-of-Thought Reasoning in Multimodal Large Language Models for Audio Generation and Editing, NeurIPS 2025, Huadai Liu, et.al.

[6]. Diff-Foley: Synchronized Video-to-Audio Synthesis with Latent Diffusion Models, NeurIPS 2023, Simian Luo, et.al.

[7]. Audio-Synchronized Visual Animation, ECCV 2024, Shentong Mo, et.al.

**Questions:**

See weaknesses section.

---

> ### Author Response · Authors · 2025-11-21
>
> We sincerely thank the reviewer for the positive assessment and for recognizing the promise of our unified architecture and the novelty of our evaluation prototypes. We deeply appreciate the detailed list of references and suggestions regarding baselines and temporal alignment metrics.
>
> ### **W1 & W2: Missing References and Additional Baselines**
>
> We commit to citing all these works and incorporating them into the related work section of our final manuscript version to provide a more comprehensive literature review.
> Furthermore, we conducted additional evaluations to compare AudioX against the suggested open-source baselines.
>
> Regarding the metrics, we have expanded our evaluation suite to include Align Acc [6]. With respect to AlignSync [7], we clarify that this metric is originally designed for *Audio-to-Video* tasks to evaluate generated video frames using a reference video. For our *Video-to-Audio* task, we utilized the AVSync Score from the same official repository, which directly measures the synchronization confidence between the input video and the generated audio.
>
> The comprehensive comparison on the VGGSound (V2A) test set is presented below:
>
> | Method | KL $\downarrow$ | IS $\uparrow$ | FD $\downarrow$ | FAD $\downarrow$ | PC $\uparrow$ | PQ $\uparrow$ | Align. $\uparrow$ | Align Acc $\uparrow$ | AVSync $\uparrow$ |
> | :--- | :---: | :---: | :---: | :---: | :---: | :---: | :---: | :---: | :---: |
> | VATT [1] | **1.40** | 10.02 | 11.71 | 2.55 | **3.64** | 5.85 | 0.25 |0.84| -0.66|
> | VAB [2] | 2.30 | 8.15 | 20.21| 3.05 | 3.52 | 5.93 | 0.24 | 0.82 | -0.64 |
> | ThinkSound [5] | 1.63 | 10.56 | 8.15 | **1.28** | 3.50 | 6.18 | 0.24 | 0.86| -0.57 |
> | MMAudio | 1.97| **14.95** | **6.18** | 2.04 | 3.38 | 5.91 | **0.35** | 0.87 | **-0.56** |
> | AudioX | 2.21 | 12.60 | 7.84 | **1.28** | 3.49 | **6.21** | 0.26 | **0.90** | -0.60 |
>
> Additionally, we evaluated AudioX on the Kling-Audio-Eval benchmark proposed in [4]. We note that [4] is a technical report and does not document quantitative results on this specific benchmark. Consequently, we report our model's performance solely to serve as a reference for future research:
>
> | Dataset | Method | KL $\downarrow$ | IS $\uparrow$ | FAD $\downarrow$ | PC $\uparrow$ | PQ $\uparrow$ | AVSync $\uparrow$ |
> | :--- | :--- | :---: | :---: | :---: | :---: | :---: | :---: |
> | Kling-Audio-Eval [4] | AudioX | 2.43 | 8.20 | 3.57 | 3.55 | 6.19 | -0.67 |
>
> The results demonstrate that AudioX achieves state-of-the-art performance in audio fidelity (lowest FAD) and production quality (highest PQ), while maintaining competitive temporal synchronization capabilities compared to specialist baselines.

---

> > ### Comment · Reviewer_CYKM · 2025-11-26
> >
> > Thanks authors for providing the additional results, I have no other concerns and will keep my rating as accept.

---

### Official Review · Reviewer_uvVZ · 2025-11-01

**Soundness:** 2
**Presentation:** 3
**Contribution:** 2
**Rating:** 4
**Confidence:** 4

**Summary:**

The paper proposes a unified any-to-audio framework based on diffusion transformers. The main contributions of the paper are:
1) large-scale dataset. 2) a unified framework for any-to-audio and 3) a proposed benchmark for instruction-followed in audio generation. The dataset is constructed by annotating the audio from the video dataset, while the proposed benchmark is constructed to test the instruction following of the model under controlled settings, and the model is proposed to use a Multimodal adaptive fusion module (MAF), which processes the input conditions before feeding it to the DiT.

**Strengths:**

- The paper is well written and easy to follow
- The model weights and proposed dataset and benchmark are promised to be released, which will make a good contribution to the field
- The direction of having a unified framework that combines multiple conditions is important.
- Extensive experiments are provided on multiple tasks.
- The proposed benchmark is solid.

**Weaknesses:**

Dataset contribution:
- The construction pipeline of the dataset is not convincing: how are the videos selected? The authors just mentioned using "existing video datasets". Were the audio-video alignment considered in the selected samples? Also, the authors mentioned that they used Gemini 2.5 to provide general captions, then Qwen2-Audio for augmentation in order to save compute. I am not convinced why Gemini 2.5 cannot provide the complete augmented instruction. The two-stage process is not well-justiied.
- The dataset quality is not well-tested. I could not find any analysis on the text-audio or audio-video alignment or the dataset in general in the paper or the supplement. Without this, it is hard to understand the dataset contributions or their comparison to existing datasets beyond scale (e.g AVSync15, AutoRecap, or even VGGSounds)

Model contribution:
- While the motivation of having an any-to-audio generation framework is interesting, it is not well justified in the paper experiments. Main experiments support text, audio, and/or video conditioning, which is already supported by existing frameworks (e.g MMAudio), except for the audio in-painting, making the framework not appear as truly any-to-audio framework and just another baseline for T2A, V2A.
- The architectural contribution is limited to the MAF module to process multiple conditions. However, it is not discussed well. For example, how is the positional encoding handled in this module? How is the temporal alignment between different conditions handled?  How does it compare to existing conditioning mechanism? (e.g MMAudio style).
- Other components of MAF are also not well justified. The gating is not intuitive and does not seem to bring significant improvement in Table 4

In general, while the paper presents significant experiments at a large scale, most of the author's choices are not well justified, and the relation to existing work (e.g, existing architecture and datasets) is not well discussed.

**Questions:**

- Is the video and audio passed to Gemini or just the audio? Figure 2 suggest only audio while L169 suggests both.
- Figure 4 is confusion. z_t is not mentioned in the figure suggesting that the combination of the conditions is fed as input for DiT rather than conditioning.

---

> ### Author Response · Authors · 2025-11-21
>
> We thank Reviewer uvVZ for recognizing the importance of the unified framework and the solidity of our benchmark. We appreciate the detailed questions regarding dataset construction and model architecture, which we address below.
>
> ### **W1: Dataset Contribution**
>
> > "how are the videos selected?"
>
> We clarify that IF-caps is built upon standard datasets in the field: VGGSound and AudioSet Strong for general audio, and V2M for music generation. These datasets have been widely validated and utilized in numerous prior works, including [1-4] for the audio domain and [5-6] for the music domain, ensuring the credibility of the audio-visual source material.
>
> > "Were the audio-video alignment considered in the selected samples?"
>
> We appreciate the suggestion regarding alignment metrics; in our revision, we have calculated the average ImageBind score between video and audio tracks for the datasets used in our work to serve as a reference for alignment quality. We will use this metric for further filtering in future work.
>
> | Dataset | ImageBind Score |
> | :--- | :---: |
> | VGGSound | 0.362 |
> | AudioSet Strong | 0.352 |
> | V2M | 0.238 |
>
> > "I am not convinced why Gemini 2.5 cannot provide the complete augmented instruction. The two-stage process is not well-justiied."
>
> Regarding the pipeline, utilizing Gemini 2.5 Pro for full-scale augmentation, while technically feasible, is cost-prohibitive. Instead, we use it to generate initial high-quality annotations, which then guide the open-source Qwen2-Audio model for large-scale augmentation. This approach effectively balances quality with efficiency. The validity of this process is proven in Table 3: the full pipeline (`GeminiCap-aug`) consistently outperforms single-stage baselines across all tasks.
>
> > "The dataset quality is not well-tested."
>
> We conduct an additional ablation (added as the 'Labels' line in the revised Table 3) using labels provided by the original video dataset. The superior performance of our caption-based model confirms the high quality of our dataset compared to original labels.
>
> &nbsp;
>
> ### **W2: Model Contribution and MAF Justification**
>
> We address the concerns regarding the model contribution and architectural design below:
>
> 1. **Justification of "Any-to-Audio" Framework**:
>    Unlike existing frameworks (e.g., MMAudio) that typically specialize in single or dual-modality tasks (T2A or V2A), AudioX is designed to handle flexible combinations of three modalities (Text, Video, and Audio) simultaneously. This is not merely "another baseline" but a unified system. Furthermore, our framework unifies Sound Effect and Music generation within a framework, whereas prior works are strictly domain-specific.
> 2. **MAF Technical Details & Comparison**:
>
>    * Positional Encoding: As described in Sec. 4.1, positional information is encoded by the temporal transformer following the visual and audio encoders.
>    * Temporal Alignment: Unlike rigid frame-to-frame mapping, MAF utilizes Learnable Queries via cross-attention (Fig. 4). These queries act as experts to assess and aggregate evidence across different data streams, allowing the model to flexibly align temporal cues from video/audio features with text instructions.
>    * Comparison with MMAudio: Unlike MMAudio's MM-DiT architecture which intertwines modality alignment with denoising via joint attention at every layer, AudioX employs a lightweight MAF module (only 60M parameters) for Early Fusion. This module explicitly aligns and compresses inputs before the backbone, ensuring that heterogeneous modalities are well aligned into a coherent representation prior to the generative process.
> 3. **Gating Intuition & Effectiveness**:
>
>    * Intuition: In multimodal settings, inputs often conflict (e.g., visual noise vs. specific text instructions). The gating mechanism intuitively acts as a dynamic filter, suppressing irrelevant or conflicting modalities to reduce interference. This aligns with established multimodal strategies [7, 8].
>    * Effectiveness: The contribution is quantitatively effective. As shown in Table 4, incorporating the Gate improves the IS score from 11.66 to 11.84 and FD from 9.72 to 9.64.
>
>
> [1] Imagebind: One embedding space to bind them all. CVPR 2023.
>
> [2] MMAudio: Taming Multimodal Joint Training for High-Quality Video-to-Audio Synthesis. CVPR 2025.
>
> [3] Tell what you hear from what you see-video to audio generation through text. NeurIPS 2024.
>
> [4] Seeing and hearing: Open-domain visual-audio generation with diffusion latent aligners. CVPR 2024.
>
> [5] Vidmuse: A simple video-to-music generation framework with long-short-term modeling. CVPR 2025.
>
> [6] UniMoE-Audio: Unified Speech and Music Generation with Dynamic-Capacity MoE[J]. arXiv 2025.
>
> [7] Flamingo: a visual language model for few-shot learning, NeurIPS 2022.
>
> [8] Language modeling with gated convolutional networks. ICML 2017.

---

> ### Author Response · Authors · 2025-11-23
>
> ### **Questions: Clarification on Inputs and Figures**
>
> We thank the reviewer for raising these points and provide the following clarifications:
>
> 1. Gemini Inputs (Line 169): We clarify that Line 169 intends to describe the final output as a "comprehensive annotated video-audio clip." However, regarding the specific input for annotation, we process the audio track to balance computational cost and annotation quality. Fig. 2 accurately reflects this setup. We have revised Line 169 in the updated manuscript to remove any ambiguity.
> 2. Figure 4 Details ($z_t$): For visual clarity, we omit the noisy latent $z_t$ in Fig. 4. In our actual implementation, the unified condition embedding $H_c$ serves as the context for the DiT's Cross-Attention layers. The noisy latent $z_t$ acts as the primary input sequence to the DiT backbone, interacting with $H_c$ to guide the generation of the final audio latent. We have explicitly clarified this input-condition interaction in the revised caption of Figure 4 to ensure architectural precision.

---

> ### Comment · Reviewer_uvVZ · 2025-11-27
>
> I thank the authors for responding to my comments. However, most of my concerns still remain
>
> **Dataset Contribution**
>
> 1- Thanks for clarifying that all videos in these datasets were used without any filtering. The scale of the dataset is limited (~260k samples) without improving the visual-audio alignment compared with audio focused datasets with associated video modality (e.g. AutoRecap) and does not improve audio-video alignment. Please note that the proposed data curation pipeline considers only the audio as an input (Figure 2) and not the associated visual modality making it more of an audio dataset curation pipeline processed on top of audio datasets with associated visual input (e.g audioset).
>
> 2- No validation of the dataset quality and pipeline was provided beyond comparing it with using the original datasets label which no only serves as a baseline but also have huge train-test mismatch with the proposed benchmark, making this baseline weak results expected. Additionally, No proper discussion of the advantage of the proposed dataset curation pipeline was provided (including the datasets that I provided in my original response).  In general, the paper does not answer the questions: what is novel in the proposed dataset curation pipeline? and if the proposed pipeline is better than previous approaches.
>
> Overall, I find the dataset contribution of this work to be limited to "recaptioning" existing datasets. The only new process is splitting recaptioning into 2 stages (Gemini and Qwen) which is not well justified. The authors stated that "cost" is the justification without any cost analysis. In my opinion, since Gemini already process the audio input and generate general caption, generating extra tokens for the augmented captions should not increase the cost a lot, but I cannot confirm this without any proper cost analysis.
>
> I acknowledge the engineering efforts in recaptioning the dataset but I cannot understand the novel contribution of the dataset beyond this. Without proper analysis of the dataset quality, it also hinders its adaptation in the field.
>
> **Model contribution**
>
> 1- MMAudio and many other existing work already support Text/video to audio. Adding another modality of audio-to-audio does not transform the proposed framework from "another baseline" to a unified "Any-to-Audio", especially that audio inpainting is quite feasible with training-free methods or by adding it to existing framework.
>
> 2- The proposed framework significantly lag behind other baselines in terms of video-audio alignment (**0.26** compared with **0.35** of MMAudio) for the V2A task. Qualitatively, the audio-video temporal alignment is quite poor compared to current approaches (e.g MMAudio). Since the paper focuses on conditioning as the main contribution, this is huge disadvantage.
>
> 3- The author mentioned that "positional information is encoded by the temporal transformer following the visual and audio encoders", meaning that no special attention was paid in the design for the temporal alignment between the conditioning and generated modality beyond processing them with a joined transformer with learnable queries. It is not obvious how well this process would captures temporal alignment. The model needs to rely on the context to understand that The Nth audio token corresponds to the Mth video token, which might explains why the model performs poorly in temporal alignment (captured by V2A task in Table. 1)
>
> 4- I still find the gating mechanism to not be well-justified. It Improves the performance by merely **0.8%** in FAD and **1.5%** in IS.
>
> Overall, I also find the model contribution to be significantly limited. The model design is not well-justifed and underperform existing approaches in temporal alignment.
>
> I appreciate the authors engineering efforts but I fail to find significant technical contribution that would justify the paper acceptance. Therefore, I decided to lower my score. I understand that my score is not aligned with the other reviewers assessment and I invite the other reviewers to engage in the discussion.

---

> > ### Author Response · Authors · 2025-12-03
> >
> > We thank Reviewer uvVZ for the response. In the following, we provide detailed clarifications regarding the Dataset Contribution and Model Contribution to address the remaining concerns.
> >
> > # **Dataset Contribution**
> >
> > ### **1. Clarification on Dataset**
> >
> > > "1-Thanks for clarifying that all videos in these datasets were used without any filtering. The scale of the dataset is limited (\~260k samples) without improving the visual-audio alignment compared with audio focused datasets with associated video modality (e.g. AutoRecap) and does not improve audio-video alignment."
> >
> > We would like to clarify **two misconceptions in the reviewer's comments** regarding the dataset construction and scale. **First**, being "built upon standard datasets" does not imply that the data is used without any filtering. **Second**, the ~260k figure refers solely to the base video-audio subset. We also utilize a 5.7 million video-music subset. Building upon these foundations, we construct the IF-caps dataset, which comprises over 7 million samples in total.
> >
> > Regarding audio-visual alignment, the standard datasets we use consist of natural videos with inherent audio-visual correspondence. The reliability of this correspondence is underscored by the fact that foundational models like ImageBind[4], which serves as a standard metric for video-audio alignment, rely on VGGSound and AudioSet for training. Furthermore, these source datasets incorporate **rigorous filtering processes**: **VGGSound[1]** is constructed via a strict 4-stage pipeline including Visual Verification, Audio Verification, and Iterative Noise Filtering; **AudioSet[2]** involves large-scale Human Verification; and **V2M[3]** employs extensive Coarse and Fine-grained filtering based on audio-visual analysis. Together with the extensive validation provided by **prior works [4-9]**, these pre-existing, rigorous verification processes guarantee the high quality and alignment of the data we utilize.
> >
> > > "Please note that the proposed data curation pipeline considers only the audio as an input (Figure 2) and not the associated visual modality making it more of an audio dataset curation pipeline processed on top of audio datasets with associated visual input (e.g audioset)."
> >
> > We clarify that our pipeline operates **not** as "an audio curation pipeline", but as a method to construct a high-quality video-audio-text multimodal dataset by annotating the audio component of existing video datasets. This integration of visual, audio, and textual modalities establishes the necessary data foundation for unified 'Anything-to-Audio' models.
> >
> > Regarding the focus on audio input, for the task of Text-to-Audio generation, we prioritize the audio content because effective supervision requires the text to describe acoustic characteristics rather than potentially silent visual details. Since the source videos possess inherent audio-visual alignment, annotating the audio track effectively bridges the text with the video's sound source, providing the most direct and accurate control signals for generation.
> >
> > [1] Chen H, Xie W, Vedaldi A, et al. Vggsound: A large-scale audio-visual dataset[C]//ICASSP 2020-2020 IEEE International Conference on Acoustics, Speech and Signal Processing (ICASSP). IEEE, 2020: 721-725.
> >
> > [2] Gemmeke J F, Ellis D P W, Freedman D, et al. Audio set: An ontology and human-labeled dataset for audio events[C]//2017 IEEE international conference on acoustics, speech and signal processing (ICASSP). IEEE, 2017: 776-780.
> >
> > [3] Tian Z, Liu Z, Yuan R, et al. Vidmuse: A simple video-to-music generation framework with long-short-term modeling[C]//Proceedings of the Computer Vision and Pattern Recognition Conference. 2025: 18782-18793.
> >
> > [4] Imagebind: One embedding space to bind them all. CVPR 2023.
> >
> > [5] MMAudio: Taming Multimodal Joint Training for High-Quality Video-to-Audio Synthesis. CVPR 2025.
> >
> > [6] Tell what you hear from what you see-video to audio generation through text. NeurIPS 2024.
> >
> > [7] Seeing and hearing: Open-domain visual-audio generation with diffusion latent aligners. CVPR 2024.
> >
> > [8] Vidmuse: A simple video-to-music generation framework with long-short-term modeling. CVPR 2025.
> >
> > [9] UniMoE-Audio: Unified Speech and Music Generation with Dynamic-Capacity MoE[J]. arXiv 2025.

---

> > ### Author Response · Authors · 2025-12-03
> >
> > ### **2. Dataset Quality and Pipeline Advantages**
> >
> > > "2-No validation of the dataset quality and pipeline was provided beyond comparing it with using the original datasets label which no only serves as a baseline but also have huge train-test mismatch with the proposed benchmark, making this baseline weak results expected."
> >
> > Regarding the validation of dataset quality, we respectfully clarify that our evaluation extends beyond the comparison with original dataset labels. In Table 3 of the main paper, we already compare our pipeline against QwenCap, which utilizes captions generated directly by Qwen2-Audio. Furthermore, to address the concern about baseline strength, we conduct an additional ablation using AudioSetCaps [10], a recent dataset that also provides captions for VGGSound and AudioSet, **as shown in the updated Table 3**. The results demonstrate that the model trained on our `GeminiCap-aug` data consistently outperforms the model trained on AudioSetCaps. This strongly validates the superior quality of our dataset curation pipeline compared to existing captioning approaches.
> >
> > > "Additionally, No proper discussion of the advantage of the proposed dataset curation pipeline was provided (including the datasets that I provided in my original response). In general, the paper does not answer the questions: what is novel in the proposed dataset curation pipeline? and if the proposed pipeline is better than previous approaches."
> >
> > We clarify the distinct novelty and advantages of our pipeline by comparing it with previous approaches: AutoReCap [11] generates large-scale unstructured captions, relies on automatic transcript filtering to exclude speech and music, and focuses solely on ambient sounds. AVSync15 [12] involves meticulous manual curation for precise audio-visual synchronization but is limited to a small scale (~1.5k samples) and lacks textual descriptions. AudioSetCaps [10] employs a chain of LALMs and LLMs to optimize descriptive natural language captions yet lacks explicit structural control. In contrast, our pipeline uniquely integrates structured annotation. Instead of merely describing the scene, we leverage the reasoning capability of MLLMs to extract objective facts, specifically event categories and counts, fine-grained Sound Event Detection (SED), and temporal relations. We then augment these key structured elements to increase data diversity.
> >
> > The advantages of our proposed pipeline are empirically validated (**as shown in Table 3**):
> >
> > (1) The model trained on our data (GeminiCap-aug) achieves SOTA performance not only in audio quality but also in instruction-following metrics compared to the baseline without augmentation (GeminiCap).
> >
> > (2) Comparative experiments demonstrate that our pipeline yields superior results compared to training with captions from AudioSetCaps as well as direct Qwen2-Audio captioning (QwenCap), confirming the higher quality of our annotations.
> >
> > [10] Bai J, Liu H, Wang M, et al. Audiosetcaps: An enriched audio-caption dataset using automated generation pipeline with large audio and language models[J]. IEEE Transactions on Audio, Speech and Language Processing, 2025.
> >
> > [11] Haji-Ali M, Menapace W, Siarohin A, et al. Taming data and transformers for audio generation[J]. arXiv preprint arXiv:2406.19388, 2024.
> >
> > [12] Zhang L, Mo S, Zhang Y, et al. Audio-synchronized visual animation[C]//European Conference on Computer Vision. Cham: Springer Nature Switzerland, 2024: 1-18.

---

> > ### Author Response · Authors · 2025-12-03
> >
> > ### **3. Contribution and Cost Justification**
> >
> > > "Overall, I find the dataset contribution of this work to be limited to "recaptioning" existing datasets. The only new process is splitting recaptioning into 2 stages (Gemini and Qwen) which is not well justified."
> >
> > **Our contribution extends beyond the novelty of the pipeline itself**, focusing fundamentally on producing high-quality data to empower model training. The fine-grained, structured data allows the model to learn precise control signals that are challenging to consistently derive from standard unstructured captions. Building on these high-quality annotations, we pave the way for strong instruction-following audio generation capabilities. Furthermore, we identify a **cross-modal regularization effect** (as detailed in Lines 427-434), where high-quality textual supervision improves video-to-audio generation quality. This demonstrates that different modalities complement each other, reducing alignment noise and **providing insights for the future development of unified models**. We commit to open-sourcing the complete dataset to benefit the community and facilitate further research.
> >
> > > "The authors stated that "cost" is the justification without any cost analysis. In my opinion, since Gemini already process the audio input and generate general caption, generating extra tokens for the augmented captions should not increase the cost a lot, but I cannot confirm this without any proper cost analysis."
> >
> > We justify the two-stage pipeline based on cost efficiency and data quality. Employing Gemini 2.5 Pro for both stages incurs approximately 4-5 times the cost due to the massive increase in output tokens required for multiple variations. Additionally, generating augmentations via a single audio input pass risks propagating interpretation biases. Furthermore, relying exclusively on a single model reduces linguistic diversity, as the outputs tend to be constrained by that model's specific generation patterns rather than offering robust, independent variations derived from structured facts.

---

> > ### Author Response · Authors · 2025-12-03
> >
> > # **Model contribution**
> >
> > ### **1. Justification of "Any-to-Audio" Framework**
> >
> > > "1- MMAudio and many other existing work already support Text/video to audio. Adding another modality of audio-to-audio does not transform the proposed framework from "another baseline" to a unified "Any-to-Audio", especially that audio inpainting is quite feasible with training-free methods or by adding it to existing framework."
> >
> > We clarify the distinct contribution of our framework compared to existing models like MMAudio and FoleyCrafter, and justify the necessity of unified training for audio-to-audio tasks:
> >
> > **Architectural Distinction and Flexibility:** Unlike MMAudio, which relies on an MM-DiT architecture that intertwines modalities via joint attention at every layer, or FoleyCrafter, which adds a semantic adapter and temporal controller to a frozen text-to-audio model, AudioX utilizes a lightweight MAF module combined with a standard DiT backbone. As detailed in the paper, MAF adaptively weights and aligns inputs via gating and query-based fusion. Our design allows AudioX to support flexible input combinations including visual, text, and audio modalities natively during training. Additionally, unlike prior works focused on sound effects, AudioX unifies both sound effect and music output domains.
> >
> > **Necessity of Unified Audio-to-Audio Training:** Audio inpainting and completion are critical for real-world editing workflows (e.g., extending a music track or repairing corrupted audio). While training-free inpainting is feasible, it often suffers from boundary inconsistency and lacks semantic guidance from other modalities, such as using video to guide the missing audio. Our end-to-end unified training explicitly optimizes the model to understand context. This superiority is empirically proven in Table A.4, where AudioX outperforms training-free baselines in both fidelity and coherence. Furthermore, our unified training reveals a new insight: multi-modal conditioning via video and text actively improves inpainting performance, demonstrating that modalities mutually reinforce each other. This benefit is not effectively attainable by simple training-free methods.
> >
> > Collectively, these structural and functional advantages distinguish AudioX from task-specific baselines, establishing it as a unified framework with superior flexibility and broad domain coverage.

---

> > ### Author Response · Authors · 2025-12-03
> >
> > ### **2. Clarification on V2A Performance**
> >
> > > "2- The proposed framework significantly lag behind other baselines in terms of video-audio alignment (**0.26** compared with **0.35** of MMAudio) for the V2A task. Qualitatively, the audio-video temporal alignment is quite poor compared to current approaches (e.g MMAudio). Since the paper focuses on conditioning as the main contribution, this is huge disadvantage."
> >
> > We clarify **a misunderstanding regarding the metrics in the reviewer's comments.** The "Align" score in Table 1 refers to the ImageBind AV score (Line 328), which measures semantic similarity, **not temporal alignment as interpreted in the comment.** To explicitly address the concern regarding temporal synchronization, we included specific temporal alignment metrics in our response to Reviewer CYKM. Notably, our model achieves the best performance in terms of Align Acc, demonstrating strong temporal synchronization capabilities.
> >
> > We also emphasize that assessing a model's capabilities should not rely solely on a single metric for one specific task. Our primary objective is to establish a unified 'Anything-to-Audio' framework that handles flexible input-output combinations and overcomes the limitations of constrained output domains. Despite this broad scope, our model remains highly comparable to baselines on the V2A task, outperforming MMAudio in four out of nine metrics (as shown in the results for Reviewer CYKM). Furthermore, AudioX achieves SOTA performance on the T2A task and significantly surpasses other baselines, including MMAudio, in instruction-following T2A ability. Our model also supports a wider range of tasks and output domains, achieving SOTA results on T2M, V2M, and TV2M tasks, which MMAudio does not support.

---

> > ### Author Response · Authors · 2025-12-03
> >
> > ### **3. Clarification on Temporal Alignment Design and Validation**
> >
> > > "3- The author mentioned that "positional information is encoded by the temporal transformer following the visual and audio encoders", meaning that no special attention was paid in the design for the temporal alignment between the conditioning and generated modality beyond processing them with a joined transformer with learnable queries. It is not obvious how well this process would captures temporal alignment. The model needs to rely on the context to understand that The Nth audio token corresponds to the Mth video token, which might explains why the model performs poorly in temporal alignment (captured by V2A task in Table. 1)"
> >
> > We clarify that the temporal transformer is explicitly designed to model temporal dynamics by encoding positional relationships into the feature sequence. The subsequent MAF module utilizes cross-attention, where learnable queries dynamically attend to these temporally encoded features, effectively learning the mapping between audio and video tokens. To empirically validate this design, we conduct an ablation study on the temporal transformer. The results demonstrate that removing this module leads to a degradation in temporal alignment metrics (e.g., a decrease in Align Acc), validating the effectiveness of this component for precise temporal synchronization.
> >
> > | Setting | KL $\downarrow$ | IS $\uparrow$ | FD $\downarrow$ | Align Acc $\uparrow$ |
> > | :--- | :---: | :---: | :---: | :---: |
> > | w/o Temporal Trans. | 2.28 | 11.54 | 8.47 | 0.85 |
> > | **Full Model** | 2.14 | 11.79 | 7.95 | 0.87 |
> >
> > Regarding the observation that the model "performs poorly in temporal alignment," we **first** reiterate the clarification from Response above: the "Align" metric in Table 1 refers to the ImageBind AV score, which measures semantic similarity, not temporal synchronization. **Second,** when evaluated on proper temporal alignment metrics (such as Align Acc provided in our response to Reviewer CYKM), our model achieves strong performance (0.91), comparable to or surpassing strong baselines like MMAudio. **Furthermore,** qualitative samples on our anonymous demo page demonstrate precise synchronization. Together, these results validate the effectiveness of our architecture in capturing temporal dynamics.

---

> > ### Author Response · Authors · 2025-12-03
> >
> > ### **4. Gating Mechanism Justification**
> >
> > > "I still find the gating mechanism to not be well-justified. It Improves the performance by merely **0.8%** in FAD and **1.5%** in IS."
> >
> > We justify the gating mechanism by emphasizing the holistic effectiveness of the MAF module. The Gating mechanism functions in conjunction with the Query module; evaluating it in isolation underestimates its role. As shown in Table 4, the Full MAF (Gate + Query) achieves a substantial performance boost compared to the baseline without MAF (IS: 10.70 to 11.84, FD: 11.60 to 9.64). Within this system, the Gate acts as an essential pre-filter to calibrate signals for the Query mechanism, ensuring the stability and superior quality of the final generation.

---

### Official Review · Reviewer_azZ7 · 2025-11-01

**Soundness:** 4
**Presentation:** 3
**Contribution:** 4
**Rating:** 10
**Confidence:** 3

**Summary:**

This paper proposes AudioX, a unified "anything-to-audio" generation framework to resolve the lack of unified frameworks for handling diverse input modalities generation.  It also provides the  IF-caps dataset to resolve the scarcity of large-scale, high-quality multimodal training data. Experiments show that AudioX enables high-fidelity audio/music generation from flexible inputs while enhancing instruction-following capabilities.

**Strengths:**

1. Audiox present a unified framework to support audio/music generation from text, video, image, and audio inputs in a single model, breaking modality/domain constraints of specialist models.
2. The authors build a large-scale Dataset, IF-caps, provides structured, fine-grained multimodal supervision, solving data scarcity for unified training.
3. The authors plans to release code, model, and IF-caps to support reproducibility

**Weaknesses:**

1. The paper’s experiments and dataset design are focused on short audio clips (mostly 10 seconds), with no discussion or evaluation of long-form audio generation (e.g., clips longer than 30 seconds. For long-form generation, there are many aspects, including temporal coherence, memory and computational efficiency, error acccumulation should be evaluated. Without them, the paper cannot fully demonstrate AudioX’s practical value for real-world applications that require extended audio.
2. The experiment only compares 4 methods (Caption2Audio, Im2Wav, Seeing&Hearing, AudioX) on a single set of metrics (KL, IS, FD, FAD, Align.), using "the same settings as in (Xing et al., 2024)" () but without specifying:
3. The paper does not provide any strategies to reduce computational cost to make it reproducible for small labs.

**Questions:**

1. Whether the images include "rare modalities" (e.g., underwater scenes, ancient artifacts, abstract art) that are less frequent in training data.
2. Whether the image-to-audio generation relies on human-written prompts, and whether rare objects/scenes were tested.

---

> ### Author Response · Authors · 2025-11-21
>
> We are deeply encouraged by the reviewer's recognition of AudioX as a "strong accept" and for highlighting the value of our unified framework! We appreciate your insightful comments and provide the following clarifications to further strengthen the manuscript.
>
> ### **W1: Limitations on Long-form Audio Generation**
>
> We acknowledge that end-to-end long-form generation remains a challenge in the field. Similar to most strong baselines (e.g., MMAudio, Tango, AudioLDM family, FoleyCrafter), our current work primarily focuses on the 10-second clip generation to ensure maximum acoustic fidelity and semantic alignment.
>
> Despite this focus, our model is capable of generating audio longer than 10 seconds through a sliding window and completion strategy. As demonstrated in our demo page (specifically the first Video-to-Music sample and Comparison V2M sample 2), AudioX can generate coherent long-form music. We appreciate the reviewer's insight regarding temporal coherence and error accumulation. While we view our current unified framework as a foundational step, solving these long-form specific challenges efficiently is a key priority for our future work.
>
> &nbsp;
> ### **W3: Computational Cost and Efficiency**
>
> We thank the reviewer for the suggestion regarding accessibility. To support the community and small laboratories, we commit to open-sourcing our model weights and code, serving as a solid foundation for diverse downstream tasks. During inference, AudioX is relatively resource-friendly, requiring only approximately 10.3 GB of VRAM, making it accessible on consumer-grade GPUs.
>
> While our primary objective in this work is to establish a unified framework and maximize generation quality, we recognize the importance of efficiency. To evaluate this, we benchmark the inference speed of AudioX against representative Text-to-Audio baselines on a single NVIDIA H800 GPU.
>
> To ensure a fair comparison, all models are configured to perform 50 inference steps. We measure the average inference time required to generate a 10-second audio sample over 1,000 runs following a warm-up phase.
>
> | Method | Inference Time (s) | Peak VRAM (GB) |
> | :--- | :---: | :---: |
> | AudioLDM | 0.89 | 2.9 |
> | AudioLDM-2 | 3.73 | 4.1 |
> | Stable-Audio-Open | 1.81 | 14.1 |
> | AudioX | 1.75 | 10.3 |
>
> The results demonstrate that AudioX achieves comparable inference speeds with moderate VRAM usage. This confirms that our unified framework maintains competitive efficiency while offering superior instruction-following capabilities. We plan to further optimize the model's efficiency in future iterations.
>
>
>
> &nbsp;
>
> ### **W2&Q: Clarification on Image-to-Audio generation**
>
> ***Test Setting:*** To address the reviewer's questions regarding our zero-shot image-to-audio capabilities, we provide the following details: We achieve zero-shot generation by padding static images to simulate video sequences. We followed the exact experimental protocol of [1], evaluating on 3k clips from the VGGSound test set. Consistent with their setting, we processed the keyframes using AnimeGANv2 [2] to transfer them into "Paprika style" before feeding them to the model. Metrics KL, IS, FD, and FAD are defined in Appendix A.2, while "Align." measures the ImageBind Score between the input image and the generated audio.
>
> ***Rare Modalities & Objects:*** We carefully inspected this test set and confirmed that it encompasses diverse and "rare" modalities. Specific examples include underwater scenes (e.g., scuba diving) and historical architecture (e.g., church bells). Moreover, the successful handling of style-transferred inputs further validates the model's robustness to abstract visual representations. We have updated the demo page with image-to-audio samples.
>
> ***Independence from Human Prompts:*** Our model does **not** rely on human-written descriptions of the content to generate semantically aligned audio. While the model supports detailed text control, for this zero-shot evaluation, we used only generic, flexible instructions such as *"Generate general audio for the image"*, *"Synthesize audio based on the visual content"*, *"Create audio that aligns with this image"*, and *"Craft a short sound effect that matches the image"*.
>
> [1] Xing Y, et al. Seeing and hearing: Open-domain visual-audio generation with diffusion latent aligners[C]//Proceedings of the IEEE/CVF Conference on Computer Vision and Pattern Recognition. 2024: 7151-7161.
>
> [2] Xin Chen. Animeganv2, https://github.com/TachibanaYoshino/AnimeGANv2/, 2022

---

### Official Review · Reviewer_iAvY · 2025-11-11

**Soundness:** 3
**Presentation:** 3
**Contribution:** 3
**Rating:** 6
**Confidence:** 4

**Summary:**

This paper presents AudioX, a unified framework for "anything-to-audio" generation, designed to produce high-quality audio (both sound effects and music) from a flexible combination of multimodal inputs, including text, video, and audio signals. The work makes four primary contributions:
- AudioX Framework: A unified model built on a Diffusion Transformer (DiT) backbone.
- MAF Module: A novel "Multimodal Adaptive Fusion" (MAF) module that adaptively fuses these diverse input modalities to reduce interference and improve conditioning.
- IF-caps Dataset: A new, large-scale (7 million sample) dataset for multimodal audio generation. This dataset is not manually labeled but created via a structured data annotation pipeline that uses a powerful MLLM (Gemini 2.5 Pro) for high-quality initial annotations, which are then augmented at scale by another model (Qwen2-Audio).
- T2A-bench: A new benchmark designed specifically to evaluate fine-grained, instruction-following capabilities in text-to-audio models, such as event counting and temporal ordering.

**Strengths:**

- Unified Generalist Model: The paper successfully presents a single generalist model, AudioX, that performs competitively across a very wide array of "anything-to-audio" tasks (T2A, T2M, V2A, V2M, inpainting, etc.).
- Superior Controllability: The model's key strength is its SOTA performance on instruction-following benchmarks (T2A-bench and AudioTime), demonstrating a new level of fine-grained control over audio content, such as event ordering and counting .
- Novel Data Pipeline: The IF-caps dataset and its generation pipeline are a significant contribution, providing a practical blueprint for overcoming data scarcity by using LLMs for high-quality annotation and augmentation.
- Strong Ablation Studies: The paper includes excellent ablation studies that validate both the architectural contribution (the MAF module, Table 4) and the data contribution (the IF-caps pipeline, Table 3).

**Weaknesses:**

- Dependency on Proprietary Models: The entire 7-million-sample IF-caps dataset is bootstrapped from annotations generated by Gemini 2.5 Pro. This makes the data curation process heavily reliant on a closed-source, proprietary, and expensive black-box model, raising significant concerns about reproducibility.
- Potential Evaluation Bias: The new T2A-bench is also evaluated using Gemini 2.5 Pro as the "automated judge". While the paper describes a thoughtful two-step "blind annotation" pipeline to mitigate bias , this is still a form of LLM-as-a-judge. Using the same model family to both generate the training data (via IF-caps) and judge the evaluation (T2A-bench) creates a potential for self-referential bias that is not fully addressed.
- Overclaim on "Image" Modality: The abstract and introduction repeatedly claim the framework is for "text, video, image, and audio". However, "image" is absent from the main model architecture (Fig. 4), training process (Sec 4.2), and main results tables (Table 1, 2). It only appears as a zero-shot task in the appendix (Table A.6). This is an overstatement of the model's core, trained capabilities.

**Questions:**

- The data generation (IF-caps) and the new benchmark (T2A-bench) both rely heavily on Gemini 2.5 Pro. Can the authors comment on the risk of this self-referential loop, where a model trained on Gemini-annotated data is then shown to be SOTA on a benchmark judged by the same Gemini model?
- The T2A-bench's automated evaluation pipeline relies on Gemini 2.5 Pro's ability to perform blind annotation (SED, counting, etc.) . How was the judge (Gemini) itself validated for accuracy on these tasks? If the judge is inaccurate, the benchmark scores in Table 2 are not reliable.
- Why is "image-to-audio" listed as a core input modality in the abstract when it is not part of the main architecture diagram and only appears as a zero-shot appendix task? Was the model ever trained on image-audio pairs?

---

> ### Author Response · Authors · 2025-11-21
>
> We sincerely thank Reviewer iAvY for recognizing AudioX as a unified generalist model with superior controllability and for acknowledging the significance of our IF-caps dataset and data pipeline. We appreciate your constructive feedback regarding evaluation bias and the image modality claim, which we address below.
>
> ### **W1: Data Reproducibility**
>
> To address the concern regarding reproducibility, we commit to open-sourcing the IF-caps dataset. This ensures the community can benefit from this high-quality data without incurring API costs. Furthermore, our data curation pipeline is designed to be a highly generalizable method. While we utilized Gemini 2.5 Pro for its state-of-the-art capabilities at the time, the pipeline remains effective with other models. With the rapid advancement of multimodal understanding, recent open-source models such as Qwen3-Omni [1] have demonstrated performance comparable to, or even exceeding, Gemini 2.5 Pro on audio understanding benchmarks. This allows future researchers to reproduce or scale our pipeline using entirely open-source solutions.
>
> &nbsp;
>
> ### **W2: Bias Mitigation**
>
> We acknowledge the potential risk of bias when using the same model family for data captioning and evaluation. However, we mitigate this risk through three key aspects. **First**, we selected Gemini 2.5 Pro as the evaluator only after extensive preliminary human testing confirmed its high reliability as an objective instruction-following judge. **Second**, our proposed two-step pipeline restricts Gemini 2.5 Pro to the role of a logic-based validator that verifies factual constraints (Count, Order, Timestamp, and Category) rather than generating subjective quality scores. **Third**, this automated evaluation is only one component of our multidimensional evaluation framework. Our conclusions are consistently supported by other evaluation dimensions in AudioTime, confirming that AudioX achieves state-of-the-art performance in instruction following.
>
> &nbsp;
>
> ### **Q1: Self-referential Loop**
>
> Regarding the specific risk of a self-referential loop, we acknowledge its theoretical possibility but argue that it is minimized because the model's cognitive role differs fundamentally between the two stages. During data curation, Gemini functions as an audio annotator, generating rich, holistic descriptions. In contrast, during evaluation, it acts strictly as an objective analyzer, extracting structured facts to verify compliance, effectively disrupting the self-referential loop. Furthermore, the high agreement with human experts (detailed in Q2) empirically proves that the "loop" has not compromised the objectivity of the evaluation.
>
> &nbsp;
>
> ### **Q2: Judge Validation**
>
> The reliability of our automated judge is supported by three distinct lines of evidence. **First**, Gemini 2.5 Pro is widely recognized as a state-of-the-art model for audio understanding, consistently topping recent benchmarks (e.g., as reported in the Qwen3-Omni technical report [1]). **Second**, to quantitatively verify its reliability in our specific context, we manually inspect a random 20% subset of evaluation samples from both AudioX and MMAudio. We observe a **91.62%** agreement rate between the automated judge’s decisions and human expert annotations. Notably, the remaining discrepancies largely stem from the judge's strict adherence to the instructions. For instance, given the prompt 'powerful thunderclap,' the judge reject a valid thunder sound for not being sufficiently 'powerful,' whereas human experts accepted it because the core sound event (thunder) is accurate. **Third**, our metrics demonstrate strong alignment with established external benchmarks. Specifically, the performance trend measured by our Ord-acc metric (AudioX > Make-An-Audio 2 > Tango 2, as shown in Figure 1) is consistent with the results obtained using the Ordering metric on the independent AudioTime benchmark. This cross-verification confirms the validity of our evaluation pipeline.
>
> &nbsp;
>
> ### **W3 & Q3: Image Modality**
>
> We thank the reviewer for pointing this out. We clarify that we did not explicitly train on image-audio pairs but utilized static image padding to simulate video sequences to achieve generation. To address the overclaim concern, we have revised the abstract and introduction in the updated manuscript.
>
> [1] Xu J, Guo Z, Hu H, et al. Qwen3-omni technical report[J]. arXiv preprint arXiv:2509.17765, 2025.

---

### Author Response · Authors · 2025-11-21
**Appreciation for Reviewers' Insightful Feedback and Recognition**

We deeply thank all reviewers for their thoughtful and insightful feedback on our submission. The positive recognition of our work is encouraging, highlighting the importance of our unified framework, the significance of the IF-caps dataset in overcoming data scarcity, and the model's superior instruction-following capabilities.

We actively aim to address the reviewers' concerns to improve the clarity and precision of the manuscript. To support reproducibility and foster future research, we fully commit to open-sourcing our code, model weights, and the dataset, inviting the community to further validate and build upon our work.

---

### Author Response · Authors · 2025-12-03
**Summary of Key Contributions and Rebuttal**

We sincerely appreciate the efforts from the reviewers, Area Chairs, and the Conference Committee during the review process. To assist in your final decision-making, we would like to provide a concise summary of our contributions and the progress made during the rebuttal period.

## **1. Summary of Contributions**

* **Unified Framework:** We propose **AudioX**, a unified framework for anything-to-audio generation that overcomes the limitations of constrained inputs and outputs. Utilizing the Multimodal Adaptive Fusion (MAF) module, the framework flexibly handles combinations of text, video, and audio inputs to synthesize corresponding audio and music, contributing to a new insight into studying generalist models for audio generation.
* **Data Curation Pipeline and Large-Scale Dataset:** To overcome data scarcity for unified training, we design a two-stage annotation pipeline: utilizing Gemini 2.5 Pro for initial structured, high-quality annotation followed by Qwen2-Audio for scalable augmentation. Based on this, we construct IF-caps, a large-scale, high-quality dataset containing over 7 million samples with fine-grained annotations.
* **Strong Performance and Research Insights:** We conduct comprehensive experiments demonstrating AudioX's strong multi-task capabilities and superior instruction-following ability. Additionally, we validate the effectiveness of our pipeline and the superior quality of our dataset through rigorous ablations. Furthermore, our unified training reveals insights into inter-modal dynamics, including a **cross-modal regularization effect** where high-quality textual supervision improves non-textual tasks (e.g., V2A), and the mutual reinforcement between modalities in complex conditioning scenarios, offering valuable insights for future unified model research.

## **2. Summary of Rebuttal Progress**

* **Reviewer iAvY (Score: 6, Confidence: 4):**

  * **Initial Concerns:** Potential evaluation bias (Self-referential loop) and "Image-to-Audio" claims.
  * **Our Response:** We substantiate the capabilities of our captioning model and validate the automated judge against human experts, achieving a 91.62% agreement rate. We further proved the evaluation's objectivity by demonstrating alignment with external benchmarks. Additionally, we clarified the experimental setting for "zero-shot" image-to-audio generation.
* **Reviewer azZ7 (Score: 10, Confidence: 3):**

  * **Initial Concerns:** Limitations on long-form generation, computational efficiency, and image-to-audio settings.
  * **Our Response:** We outline future directions for end-to-end long-form generation while demonstrating the current capability to generate long audios via a sliding-window strategy. We provide a detailed computational analysis (requiring 10.3 GB VRAM with fast inference), confirming the model's efficiency, and clarify the image-to-audio generation protocols.
* **Reviewer uvVZ (Score: 4, Confidence: 4):**

  * **Initial Concerns:** Dataset construction pipeline (video selection, alignment), validation of dataset quality, and justification of the model architecture.
  * **Our Response:** We clarify that IF-caps is built upon standard, widely validated datasets (VGGSound, AudioSet, V2M). We validate the effectiveness of the two-stage pipeline empirically (Table 3) and demonstrate dataset quality through additional ablations, showing our caption-based model outperforms label-based training. Regarding the model, we highlight AudioX's unique ability to handle flexible three-modality inputs and unify sound/music generation, providing experimental results to prove the architectural effectiveness.
  * **Discussion Phase:** The reviewer raises further concerns regarding Dataset and Model contribution.
  * **Our Clarification:**
    * **Dataset:** We correct critical misconceptions regarding dataset construction and scale. We justify the pipeline's focus on audio input based on inherent source alignment, experimentally validate the superior dataset quality and pipeline advantages over previous approaches (e.g., AudioSetCaps), and justify the contribution and cost efficiency of the two-stage approach.
    * **Model:** We clarify the architectural distinctions from baselines, correct misunderstandings regarding V2A metrics to highlight our strong temporal alignment performance, and justify the necessity of the Gating mechanism.


* **Reviewer CYKM (Score: 8, Confidence: 5):**

  * **Initial Concerns:** Missing baselines and temporal alignment metrics.
  * **Our Response:** We incorporate comparisons with VATT, VAB, and ThinkSound, and report Align Acc and AVSync metrics. Results show AudioX achieves SOTA performance in audio fidelity (lowest FAD) and strong temporal alignment (0.90 Align Acc).
  * **Reviewer Response:** Following these updates, the reviewer confirms: "keep my rating as accept."

---

### Meta-Review · Area_Chair_eM9r · 2026-01-08

**Summary:**

The paper proposes AudioX, a unified "anything-to-audio" DiT-based framework, supported by the large-scale IF-caps dataset and the T2A-bench for instruction following. Reviewers praise the model's versatility across multimodal tasks and its strong performance in fine-grained control. Key concerns include the self-referential bias of using Gemini for both data annotation and evaluation, the overstatement of image-to-audio as a core modality, and missing comparisons with several recent SOTA baselines and temporal alignment metrics.

**Reviewer Concerns:**

The author's rebuttal addressed most of the issues pointed out by the reviewers, especially providing key clarifications for the questions raised by Reviewer uvVZ.

**Reviewer Scores:**

Reviewer uvVZ might raise the rating.

---

### Decision · Program_Chairs · 2026-01-26

Accept (Poster)